# Indisulam synergizes with palbociclib to induce senescence through inhibition of CDK2 kinase activity

**Ziva Pogacar**[1], **Jackie L. Johnson**[1], **Lenno Krenning**[2], **Giulia De Conti**[1], **Fleur Jochems**[1], **Cor Lieftink**[3], **Arno Velds**[4], **Leyma Wardak**[1], **Kelvin Groot**[1], **Arnout Schepers**[1], **Liqin Wang**[1], **Ji-Ying Song**[5], **Marieke van de Ven**[6], **Olaf van Tellingen**[7], **Rene H. Medema**[2], **Roderick L. Beijersbergen**[1,3], **Rene Bernards**[1]*, **Rodrigo Leite de Oliveira**[1¤]*

1 Division of Molecular Carcinogenesis, Oncode Institute, The Netherlands Cancer Institute, Amsterdam, The Netherlands, 2 Division of Cell Biology, Oncode Institute, The Netherlands Cancer Institute, Amsterdam, The Netherlands, 3 The NKI Robotics and Screening Center, The Netherlands Cancer Institute, Amsterdam, The Netherlands, 4 Genomics Core Facility, The Netherlands Cancer Institute, Amsterdam, The Netherlands, 5 Division of Animal Pathology, The Netherlands Cancer Institute, Amsterdam, The Netherlands, 6 Mouse Clinic for Cancer and Aging, Netherlands Cancer Institute, Amsterdam, The Netherlands, 7 Division of Pharmacology, The Netherlands Cancer Institute, Amsterdam, The Netherlands

¤ Current Address: CRISPR Expertise Center, Cancer Center Amsterdam, Amsterdam University Medical Center, Amsterdam, The Netherlands

* r.bernards@nki.nl (RB); r.ld.oliveira@amsterdamumc.nl (RLO)

## Abstract

Inducing senescence in cancer cells is emerging as a new therapeutic strategy. In order to find ways to enhance senescence induction by palbociclib, a CDK4/6 inhibitor approved for treatment of metastatic breast cancer, we performed functional genetic screens in palbociclib-resistant cells. Using this approach, we found that loss of *CDK2* results in strong senescence induction in palbociclib-treated cells. Treatment with the CDK2 inhibitor indisulam, which phenocopies genetic CDK2 inactivation, led to sustained senescence induction when combined with palbociclib in various cell lines and lung cancer xenografts. Treating cells with indisulam led to downregulation of cyclin H, which prevented CDK2 activation. Combined treatment with palbociclib and indisulam induced a senescence program and sensitized cells to senolytic therapy. Our data indicate that inhibition of CDK2 through indisulam treatment can enhance senescence induction by CDK4/6 inhibition.

## Introduction

Cellular senescence is a stable cell cycle arrest and can be induced by a variety of stressors, including cancer therapies (referred to as therapy induced senescence [TIS]) [1]. Senescence is characterized by changes in cellular physiology, such as changes of cell morphology, changes in gene expression and metabolism, and secretion of a variety of proteins (collectively referred to as the senescence associated secretory phenotype [SASP]) [2]. Induction of senescence as an anti-cancer treatment can be advantageous in the short term because cell proliferation is halted

**Data Availability Statement:** All raw RNA-sequencing data are available through the GEO database under accession numbers: GSE197600,

GSE197601, GSE197602, GSE197603, and GSE197604.

**Funding:** EC | H2020 | H2020 Priority Excellent Science | H2020 European Research Council (ERC):Ziva Pogacar,Jackie L. Johnson,Giulia De Conti,Cor Lieftink,Leyma Wardak,Fleur Jochems, Kelvin Groot,Arnout Schepers,Liqin Wang,Roderick Beijersbergen,Rene R Bernards,Rodrigo Leite de Oliveira 787925 Other authors worked without finding.

**Competing interests:** R.B is the founder of the company Oncosence (https://www.oncosence. com), which aims to develop senescence-inducing and senolytic compounds to treat cancer. This does not alter our adherence to PLOS ONE policies on sharing data and materials.

and immune cells are recruited through the SASP. However, in the long term, persistence of senescent cancer cells can lead to chronic inflammation, tumor progression and migration [3]. We postulated that a "one-two punch" approach to cancer therapy, in which a first drug induces senescence and the second drug either targets the senescent cancer cells for death (senolysis) or enhances the efficacy of the immune infiltrate may be an effective anti-cancer strategy [4, 5].

Several cancer treatments have been shown to induce senescence, including chemotherapeutics and targeted agents (reviewed in [6]). For instance, targeting CDK4 and 6 with inhibitors (such as palbociclib, ribociclib and abemaciclib) induced senescence in various cancer models [7–11]. CDK4/6 are important kinases in the cell cycle, regulating the transition from G1 to S phase by phosphorylating and partially inactivating the retinoblastoma protein RB. Upon subsequent further phosphorylation of RB by CDK2, RB is functionally fully inactivated, leading to complete de-repression of E2F transcriptional activity and entry into S phase [12]. Since the majority of cancer cells have an intact *RB1* gene and thus depend on CDK4/6 kinase activity for sustained proliferation, CDK4/6 emerged as a potential target for cancer therapy. CDK4/6 inhibitors have been approved for treatment of hormone receptor positive (HR+) and human epidermal growth factor receptor 2 (HER2) negative (HER2-) breast cancer in combination with anti-hormonal therapy [13–15].

Due to the efficacy, safety and tolerability of the CDK4/6 inhibitors in HR+ breast cancer, and multiple nodes of oncogenic signals converging on CDK4/6 in multiple cancer types [16], there has been significant interest in extending their use to other cancer types. Several clinical trials using CDK4/6 inhibitors in various cancer types, such as non-small cell lung cancer, ovarian cancer and triple negative breast cancer were recently completed [17–20]. However, translating the use of CDK4/6 inhibitors to other tumor types has proven to be challenging, due to limited senescence induction and intrinsic resistance [21–23]. Better understanding of the limitations of senescence induction by CDK4/6 inhibitors may help in broadening the clinical utility of this class of cancer therapeutics.

A potential solution is using CDK4/6 inhibitors in a rational combination. For example, combining CDK4/6 inhibitors with PI3K inhibition was effective in multiple preclinical models [24, 25]. Furthermore, combining CDK4/6 inhibition with the MEK inhibitor trametinib was shown to increase senescence induction in lung cancer and colorectal cancer cells [26, 27].

Indisulam was originally identified as a sulfonamide with anticancer effects that acts as an indirect CDK2 inhibitor [28]. However, after various phase 1 and 2 clinical trials showed a response and stable disease in only 17–36% of patients, development was halted [29–35]. On a molecular level, indisulam was recently identified as being a molecular glue, targeting the splicing factor RBM39 to DCAF15, a component of a ubiquitin ligase complex, leading to degradation of RBM39 [36]. Here, we identify an unexpected synergy between indisulam and palbociclib in induction of senescence in multiple cancer types. Moreover, we find that cancer cells made senescent with palbociclib and indisulam are sensitive to the senolytic agent ABT-263.

## Results

### CDK2 loss is synergistic with palbociclib in inducing senescence in triple negative breast cancer

Palbociclib has only modest cytostatic activity in triple negative breast cancer (TNBC) cell lines. To capture the heterogeneity of TNBC, we chose three independent cell lines as models for kinome-based shRNA synthetic lethality screens to identify genes whose suppression enhances the response to palbociclib (S1A Fig in S1 File). We performed synthetic lethality

screens in CAL-51, CAL-120 and HCC1806, all of which are resistant to palbociclib (Fig 1A and S1C Fig in S1 File). Depleted shRNAs were identified by deep sequencing as described previously [37]. When comparing the relative abundance of shRNAs in palbociclib-treated to untreated cells, shRNAs targeting *CDK2* were depleted in all three cell lines (Fig 1A). Furthermore, *CDK2* was the only common hit between all three screening cell lines (Fig 1B). To validate this observation, we used individual shRNAs to knock down *CDK2* in the cell lines used for the screens (S1B Fig in S1 File). Even though *CDK2* knockdown had no effect on proliferation, we observed a decrease in proliferation in *CDK2* knockdown cells treated with palbociclib (S1C, S1D Fig in S1 File). Similarly, CAL-51-*CDK2* knockout cells had no changes in proliferation, but were more sensitive to palbociclib treatment (Fig 1C–1E).

CDK2 knockout cells treated with palbociclib showed a change in morphology indicative of senescence. To better characterize these cells, we stained them for senescence-associated β-galactosidase (SA-β-gal), an established marker of senescence [38]. We observed an increase in the number of cells positive for SA-β-gal in all three TNBC cell lines when cells lacking *CDK2* were treated with palbociclib (Fig 1F and 1G and S1E, S1F Fig in S1 File).

Given that senescent cancer cells can promote inflammation and support metastasis, senolytic therapies are being developed to obliterate the senescent cells [6]. We tested the senolytic compound navitoclax (ABT-263, an inhibitor of BCL-2, BCL-xL and BCL-W) in *CDK2* knockout TNBC cells rendered senescent with palbociclib. After pre-treating sgCDK2 and control cells with palbociclib, only cells lacking CDK2 were killed upon treatment with ABT-263 (Fig 1H).

## CDK2 inhibition is synergistic with palbociclib in multiple cancer types

Due to the heterogeneity of TNBC cell lines we hypothesized that the interaction between CDK2 and palbociclib may be a general dependency and could therefore be applied broadly to other cancer types. To address this, we tested an additional TNBC cell line (SUM159) as well as lung cancer cell lines A549 and H2122 and colorectal cancer cell lines DLD-1 and RKO. We knocked out *CDK2* (Fig 2A and 2F and S2A Fig in S1 File) and observed an increased sensitivity to palbociclib in cells harboring sgCDK2 compared to control cells (Fig 2B, 2C, 2G and 2H and S2B Fig in S1 File). Furthermore, cells harboring sgCDK2 treated with palbociclib showed an increased number of SA-β-gal positive cells in SUM159 and A549 (Fig 2D, 2E, 2I and 2J) and the enlarged size and flat morphological features of cellular senescence. SUM159 CDK2 knock-out cells also showed sensitivity to the senolytic agent ABT-263 (S2C Fig in S1 File). Unequivocal identification of senescence in cancer cells can be difficult due to the lack of gold-standard markers of the senescent state. Previous studies have identified gene signatures associated with senescence [2, 39] or list of genes differentially expressed in senescence [40, 41]. We therefore used transcriptome analyses to further characterize these cells. We observed that A549 cells showed enrichment in four out of five tested senescence signatures in RNA-seq experiments when comparing treated and untreated sgCDK2 cells (Fig 2K). This shows that senescence induction upon *CDK2* loss and CDK4/6 inhibition has little context-dependency.

## Indisulam phenocopies CDK2 loss and induces senescence in combination with palbociclib

Even though the interaction between *CDK2* loss and palbociclib induces senescence in a broad panel of cell lines, the lack of a selective CDK2 inhibitor complicates further development of this concept. While a variety of compounds targeting CDK2 are available, they tend to be non-specific and also target other CDKs, such as CDK1/5/7/9 (reviewed in [42]which diminishes their utility. Furthermore, the off-target effects of these compounds often lead to toxicity and prevent their use in combination therapies. In search of a CDK2 inhibitor we came across

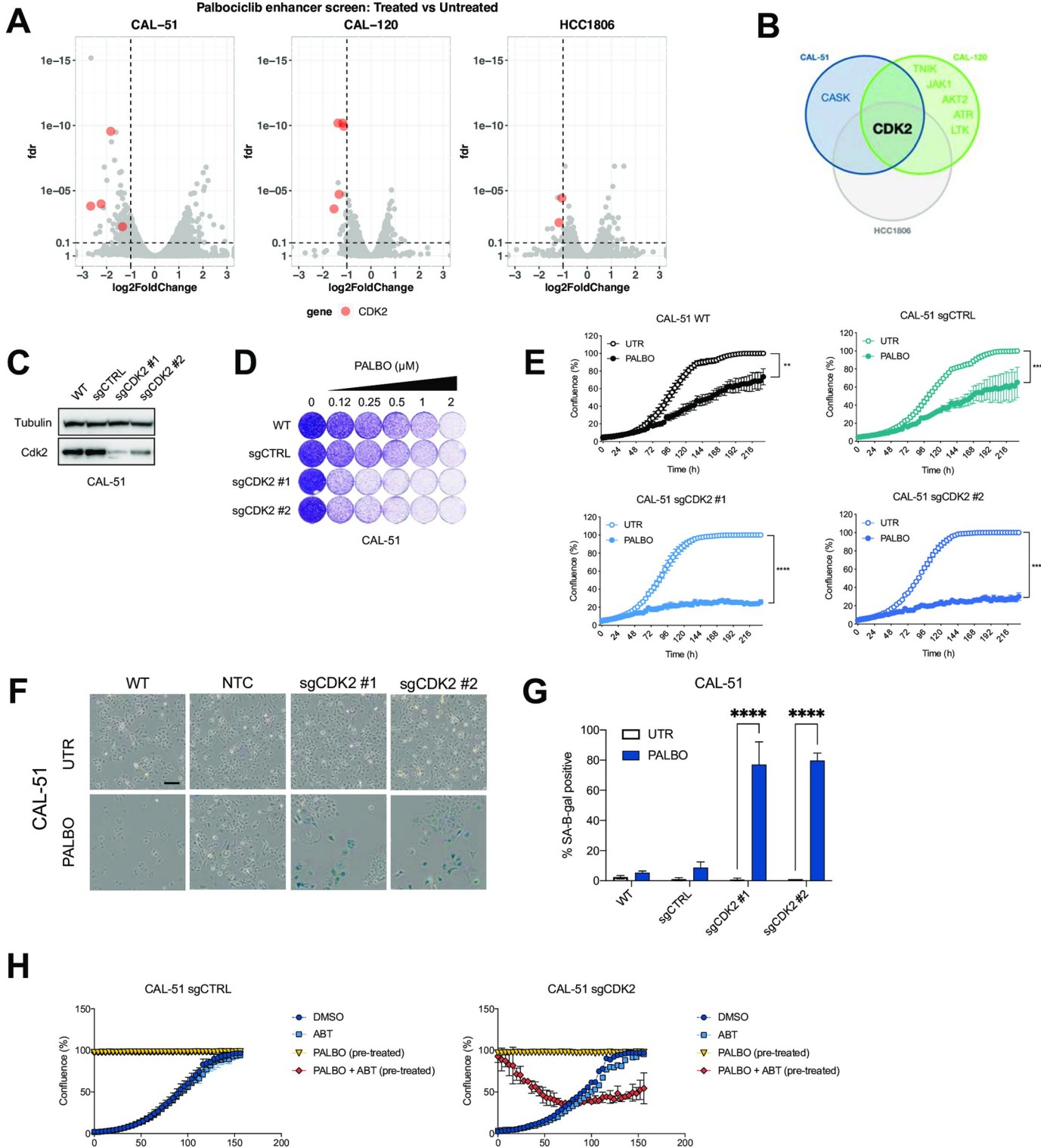

**Fig 1. CKD2 loss is synergistic with palbociclib in induction of senescence in triple negative breast cancer. A** Volcano plot of hit selection. shRNA counts of CAL-51, CAL-120 and HCC1806 were compared between palbociclib treated and untreated conditions. Each dot represents an individual shRNA. Y axis shows the false discovery rate (FDR) and X axis shows fold change between conditions. The cutoffs of 0.1 FDR and -1 log2 fold change are represented by the dashed lines. Red dots indicate shRNAs targeting CDK2. Hits were selected as genes that were represented with at least 2 independent shRNAs. **B** Venn diagram shows overlap of hits between CAL-51, CAL-120 and HCC-1806. Hits were selected as genes that were represented with at least 2 independent shRNAs. **C-G** Screen

validation: CDK2 was knocked out with two independent sgRNAs in CAL-51 cells. Polyclonal population of sgCDK2 cells was used for experiments. **C** Western blot analysis of CDK2 levels in CAL-51 sgCDK2 and control cells. Tubulin was used as loading control. Representative images of two independent experiments are shown (n = 2). **D** Long term colony formation assay of CAL-51. Wild-type, control and sgCDK2 cells were treated with indicated doses of palbociclib for 10 days. Representative of three independent experiments is shown (n = 3). **E** Proliferation assay of CAL-51. Cells were treated with 0.5 μM of palbociclib or DMSO. Mean of three technical replicates representative of two independent experiments (n = 2) is shown and error bars indicate standard deviation. The end point confluency of all conditions were analysed using two-way ANOVA with Šidák's post-hoc test (** $p < 0.01$ **** $p < 0.0001$). **F** SA-β-gal staining in CAL-51 cells treated for 10 days with 0.5 μM of palbociclib. Scale bar indicates 100 μm. Representative images of two independent experiments are shown (n = 2). **G** Quantification of SA-β-gal positive cells shown in **F**. CAL-51 cells were treated for 10 days with 0.5 μM of palbociclib. Bars represent mean ± SD of triplicates. Data was analysed using two-way ANOVA with Šidák's post-hoc test (**** $p < 0.0001$). **H** Proliferation assay of CTRL or CAL-51 sgCDK2 cells treated with senolytic drug ABT-263. Cells were pre-treated with 2 μM of palbociclib for 10 days to induce senescence, then seeded in high density (100% confluence) and treated with palbociclib or a combination of palbociclib and 5 μM ABT-263. Proliferating cells, which were not pre-treated, were seeded at low density and treated with DMSO or ABT-263. Mean of three technical replicates representative of two independent experiments (n = 2) is shown and error bars indicate standard deviation.

indisulam, a sulfonamide that was described as an indirect CDK2 inhibitor [28]. When we treated the cells with the combination of palbociclib and indisulam we observed a decrease in proliferation in all tested cell lines (Fig 3A and 3B and S3A, S3B Fig in S1 File). Furthermore, treatment with indisulam and palbociclib showed an increase in the number of cells positive for SA-β-gal (Fig 3C and 3D and S3C, S3D Fig in S1 File). Both SUM159 and A549, as well as CAL-51, DLD-1 and RKO senescent cells induced by palbociclib and indisulam were sensitive to ABT-263 (Fig 3E and S3E Fig in S1 File).

We then set out to further characterize the senescence phenotype by testing four different senescence markers by Western blot. We observed a decrease in phosphorylated RB and increase in p21 in both cell lines. There was also an increase in γH2AX, although less apparent in A549. Furthermore, there was an increase in p16INK4A in SUM159. Since A549 cells are p16 null, we examined an additional marker Lamin B1, which was reduced upon the combination treatment. We observed a reduction in CDK2 protein levels in palbociclib treated samples. However CDK2 was not further reduced upon the combination treatment samples since indisulam is an indirect CDK2 inhibitor that does not influence the protein abundance (Fig 3F and S3H Fig in S1 File).

Next, we treated A549 and SUM159 cells with indisulam, palbociclib and the combination, and performed RNA-sequencing. We observed enrichment in senescence signatures when comparing the combination-treated cells to single drugs or untreated conditions (Fig 3G). Additionally, we tested the recently developed PF-0687360 compound, which is described to inhibit CDK2/4/6 [43, 44]. Treatment with PF-0687360 led to an increase of SA-β-gal positive cells in SUM159 and A549 (S3F, S3G Fig in S1 File) as well as enrichment in senescence signatures (S3I Fig in S1 File), further validating inhibition of CDK2 with CDK4/6 as a senescence inducing combination.

## Combination of indisulam and palbociclib impairs tumor growth *in vivo*

To extend the findings to an *in vivo* model, we tested if *CDK2* loss leads to growth arrest when combined with palbociclib treatment in mice xenografts. We generated *CDK2* KO clones in A549 and CAL-51 cells and then engrafted A549 subcutaneously and CAL-51 orthotopically in NMRI nude mice. However, *CDK*2 seems to be essential for growth *in vivo* as the growth of *CDK*2 knock-out tumors was severely impaired, making genetic validation technically not feasible (S4A, S4B Fig in S1 File). We then proceeded to test the combination of palbociclib and indisulam *in vivo*. Firstly, we performed a PK/PD experiment and determined that both drugs were stable in plasma (S4C, S4D Fig in S1 File). Next, we engrafted A549 cells subcutaneously and treated the mice with vehicle, palbociclib, indisulam or the combination. We observed a reduction in tumor growth in animals treated with the drug combination compared to the single treatments (Fig 4A). Furthermore, immunohistochemical analysis showed decrease of the

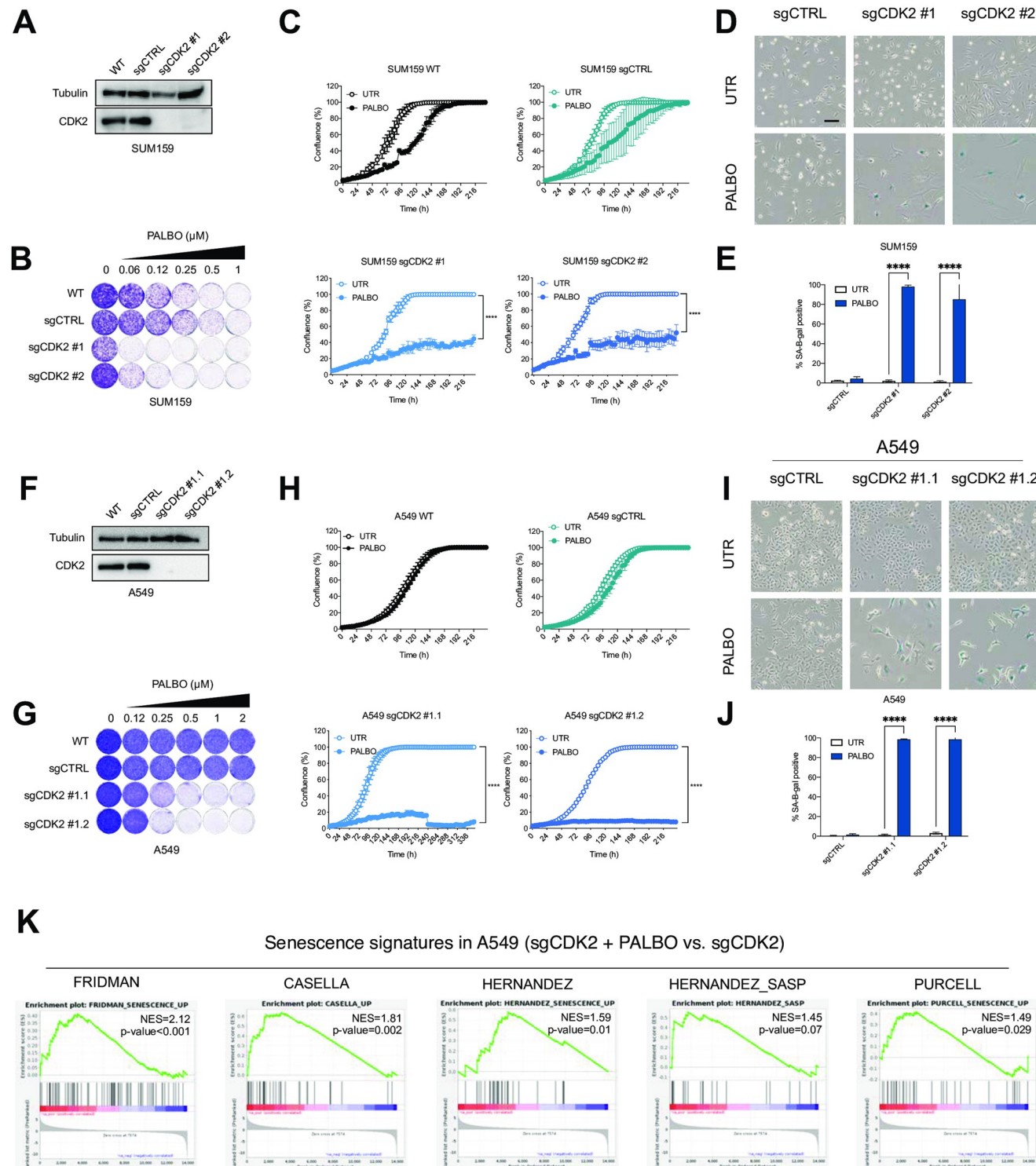

**Fig 2. CDK2 inhibition is synergistic with palbociclib in multiple cancer types. A-E** CDK2 was knocked out in SUM159 cells using two independent sgRNAs. **A** Western blot analysis of SUM159 sgCDK2 cells and control cells. Tubulin was used as a loading control. Representative images of two independent experiments are shown (n = 2). **B** Long term colony formation assay of SUM159 sgCDK2 and control cells with palbociclib was performed for 8 days. Representative of two independent experiments are shown (n = 2). **C** Proliferation assay of SUM159 sgCDK2 and control cells treated with 0.25 μM of palbociclib. Mean of three technical replicates representative of two independent experiments (n = 2) is shown and error bars indicate standard deviation.

The end point confluency of all conditions were analysed using two-way ANOVA with Šidák's post-hoc test (**** p< 0.0001). **D** SA-β-gal staining in SUM159 sgCDK2 and control cells treated with 0.25 μM of palbociclib for 8 days. Scale bar indicates 100 μm. Representative images of two independent experiments are shown (n = 2). **E** Quantification of SA-β-gal positive cells shown in **D**. SUM159 cells were treated for 8 days with 0.25 μM of palbociclib. Bars represent mean ± SD of triplicates. Data was analysed using two-way ANOVA with Šidák's post-hoc test (**** p< 0.0001). **F-J** CDK2 was knocked out in A549 and two single cell clones were selected. **F** Western blot analysis of A549 sgCDK2 cells and control cells. Tubulin was used as a loading control. Representative images of two independent experiments are shown (n = 2). **G** Long term colony formation assay of A549 sgCDK2 and control cells with palbociclib was performed for 10 days. Representative of two independent experiments are shown (n = 2). **H** Proliferation assay of A549 sgCDK2 and control cells treated with 0.5 μM of palbociclib.Mean of three technical replicates representative of two independent experiments (n = 2) is shown and error bars indicate standard deviation. The end point confluency of all conditions were analysed using two-way ANOVA with Šidák's post-hoc test (**** p< 0.0001). **I** SA-β-gal staining in A549 sgCDK2 and control cells treated with 0.5 μM of palbociclib for 10 days. Scale bar indicates 100 μm. Representative images of two independent experiments are shown (n = 2). **J** Quantification of SA-β-gal positive cells shown in **I**. A549 cells were treated for 10 days with 0.5 μM of palbociclib. Bars represent mean ± SD of triplicates. Data was analysed using two-way ANOVA with Šidák's post-hoc test (**** p< 0.0001). **K** GSEA of previously published senescence gene sets comparing A549 sgCDK2 cells treated with 2 μM palbociclib for 10 days with untreated cells. Normalized enrichment score (NES) and p-values of enrichment score are shown. The experiment was performed in duplicates.

proliferation marker Ki67 and increase of the CDK inhibitor p21 in tumors treated with the drug combination, compared to single treatments or control groups (Fig 4B and 4C). We conclude that the combination of indisulam and palbociclib is well tolerated in vivo and leads to impaired tumor growth and reduced proliferation.

## Indisulam prevents activation of CDK2 leading to cell cycle arrest when combined with palbociclib

To better understand the effects of indisulam on CDK2 we first performed an *in vitro* kinase activity assay. In short, we added indisulam to different cyclin/CDK complexes in vitro and measured the kinase activity as previously described [45]. We did not observe a direct and specific inhibition of CDK2 by indisulam, which was in line with previous reports on indisulam being an indirect CDK2 inhibitor (S5A Fig in S1 File). Recently, indisulam has been characterized as a molecular degrader, bringing together a splicing factor RBM39 with DCAF15 substrate receptor of CUL4a/b ubiquitin ligase complex [36]. This leads to ubiquitination and proteasomal degradation of RBM39, leading to accumulation of splicing errors. We therefore asked whether RBM39 degradation plays a role in the senescence induction by palbociclib and indisulam. Indeed, we observed a degradation of RBM39 in cells treated with indisulam or the combination with palbociclib (Fig 5A). To understand if the combination effect of palbociclib and indisulam is dependent on RBM39 we used shRNAs to knock down RBM39 (Fig 5B). We observed that cells with reduced RBM39 expression and treated with palbociclib showed a reduction in growth (Fig 5C) and became positive for SA-β-gal staining (Fig 5D and 5E). This suggests that the senescence induction upon indisulam and palbociclib treatment is mediated through RBM39 degradation.

To characterize splicing errors downstream of RBM39 degradation we treated A549 and SUM159 cells with indisulam, palbociclib or the combination and collected an RNA-seq data. Upon quantifying the splicing errors (see Methods) we detected an increase of splicing errors in indisulam-treated and combination-treated cells with skipped exons being the most common splicing error class detected (Fig 5F).

To elucidate how indisulam-induced splicing errors affect CDK2 activity, we made use of a CDK2 reporter construct that changes its subcellular localization depending on CDK2 activity [46]. Cells with low CDK2 activity show fluorescence in the nucleus and cells with active CDK2 show fluorescence in the cytoplasm. Upon 24 hours treatment with a matrix of increasing concentrations of indisulam, palbociclib and the combination we imaged fixed cells and determined the nuclear and cytoplasmic fluorescence signal, indicative of CDK2 activity. We observed a decrease of CDK2 activity in cells treated with the combination of indisulam and palbociclib (Fig 5G). To exclude a potential confounding effect of cell cycle arrest leading to

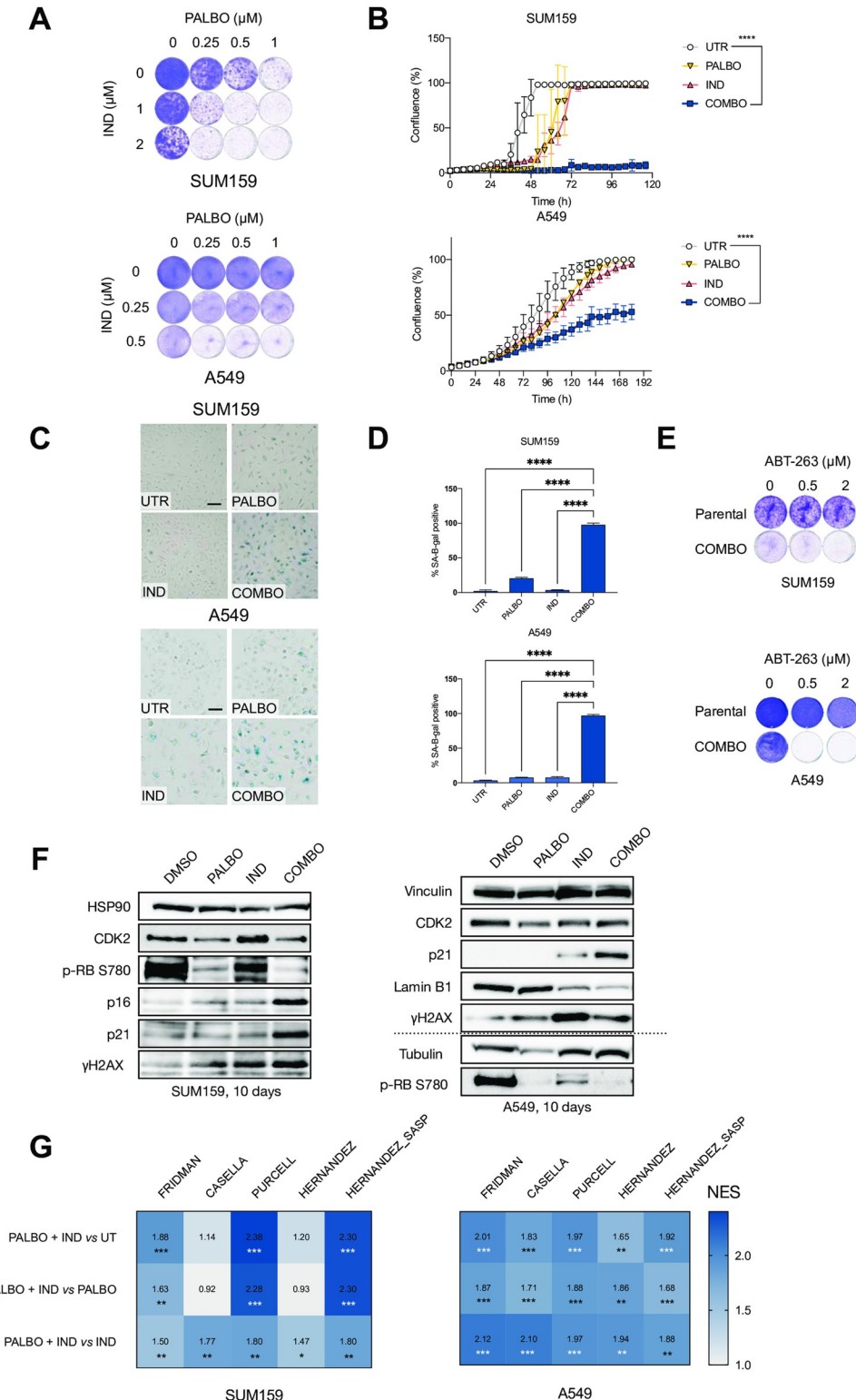

**Fig 3. Indisulam phenocopies CDK2 loss and induces senescence in combination with palbociclib. A** Long term colony formation assay of SUM159 and A549 cells treated with palbociclib, indisulam and the combination for 10 days. Representative of three independent experiments are shown (n = 3). **B** Proliferation assay of SUM159 and A549 cells treated with palbociclib, indisulam and the combination. SUM159 cells were treated with 0.5 μM of palbociclib and 2 μM indisulam and A549 with 2 μM palbociclib and 0.5 μM indisulam. Mean of three technical replicates

representative of two independent experiments (n = 2) is shown and error bars indicate standard deviation. The end point confluency of all conditions were analysed using one-way ANOVA with Dunnett's post-hoc test (**** p< 0.0001). **C** SA-β-gal staining in SUM159 and A549 cells treated with palbociclib, indisulam and combination. SUM159 were treated with 0.5 μM of palbociclib and 2 μM indisulam and A549 with 2 μM palbociclib and 0.5 μM indisulam for 10 days. Scale bar indicates 100 μm. Representative images of three independent experiments are shown (n = 3). **D** Quantification of SA-β-gal positive cells shown in **C**. SUM159 were treated with 0.5 μM of palbociclib and 2 μM indisulam and A549 with 2 μM palbociclib and 0.5 μM indisulam for 10 days. Bars represent mean ± SD of triplicates. Data was analysed using one-way ANOVA with Dunnett's post-hoc test (**** p< 0.0001). **E** Long term colony formation assay of SUM159 and A549 cells pre-treated with 0.5 μM of palbociclib plus 2 μM of indisulam for SUM159 and 2 μM of palbociclib plus 0.5 μM of indisulam for A549 for 2 weeks. Senescent and parental cells were then treated with 0.5 μM and 2 μM of ABT-263 for 1 week. Representative of two independent experiments are shown (n = 2). **F** Western blot analysis of SUM159 and A549 cells treated with palbociclib, indisulam and combination. SUM159 were treated with 0.5 μM of palbociclib and 2 μM indisulam and A549 with 2 μM palbociclib and 0.5 μM indisulam for 10 days. HSP90, vinculin and tubulin were used as a loading control. Dotted line indicates a separate experiment. Representative images of two independent experiments are shown (n = 2). **G** GSEA of previously published senescence gene sets comparing A549 and SUM159 cells treated with palbociclib, indisulam or the combination. SUM159 were treated with 0.5 μM of palbociclib and 2 μM indisulam and A549 with 2 μM palbociclib and 0.5 μM indisulam for 10 days. Comparisons of normalized enrichment scores of combination with untreated, and combination with single drugs is shown. Numbers indicate p-value (*p<0.05; **p<0.01; ***p<0.001).

reduced CDK2 activity in the combination treatment, we performed a FACS-based experiment using the concentrations of indisulam and palbociclib that showed reduced CDK2 activity in the imaging experiment.We did not observe cell cycle differences between the single treatments and the combination after 24h (Fig 5H), which indicates that CDK2 inactivation happens upstream of cell cycle arrest.

To investigate the dynamic of CDK2 inactivation upon treatment we performed a live imaging experiment using CDK2 reporter cells. We included a positive control, a CDK1/2 inhibitor SNS-032. After beginning the imaging, the drugs were added and cells were followed through their next cell cycle. We first followed the cells that showed the fluorescence signal in the cytoplasm, indicative of active CDK2. We observed inactivation of CDK2 when we added SNS-032, but not the other drugs. Next, we followed the cells that showed fluorescence in the nucleus, and therefore had low activity of CDK2. We observed that treatments with SNS-032 and palbociclib prevented CDK2 activation, as expected. Remarkably, treatment with indisulam prevented CDK2 activation as well (Fig 5I).

When examining the effects of the combination treatment on the cell cycle proteins we observed a stronger reduction in phosphorylated RB in the combination treated cells (Fig 5J and S5B Fig in S1 File). While levels of total CDK2 were only slightly reduced, the difference is likely in the levels of active CDK2. Additionally, Cyclin E levels were increased upon palbociclib treatment, which can be explained by reduced CDK2 activity regulating cyclin E levels [47, 48]. We also observed a decrease in Cyclin H, a member of the CDK activating complex (CAK). CAK phosphorylates and activates CDK2 and we hypothesized that indisulam-induced downregulation of Cyclin H could prevent CDK2 activation. To this end, we tested the expression levels of *CCNH* upon treatment with indisulam using qPCR and observed a downregulation in both A549 and SUM159 cells (Fig 5K). Additionally, cells with knockdown of RBM39 also showed a reduction in *CCNH* (Fig 5L). This might indicate that indisulam-induced *CCNH* downregulation prevents CDK2 activation through the CAK complex inactivation. We then analyzed the RNA-seq data generated in Fig 5F by performing a kinase enrichment analysis [49]. Genes associated with CDK1 and CDK2 were over-represented in the list of differentially expressed genes in A549 and SUM159 cells treated with indisulam (S5C Fig in S1 File). This further indicates that treatment with indisulam reduces the activity of CDK2. To confirm the effect of indisulam on CDK2, we overexpressed CDK2 in A549 cells using both transient transfection (Fig 5M) and stable integration of lentiviral vectors (S5D Fig in S1 File). We observed an increase in proliferation rescuing the senescence induction in cells that

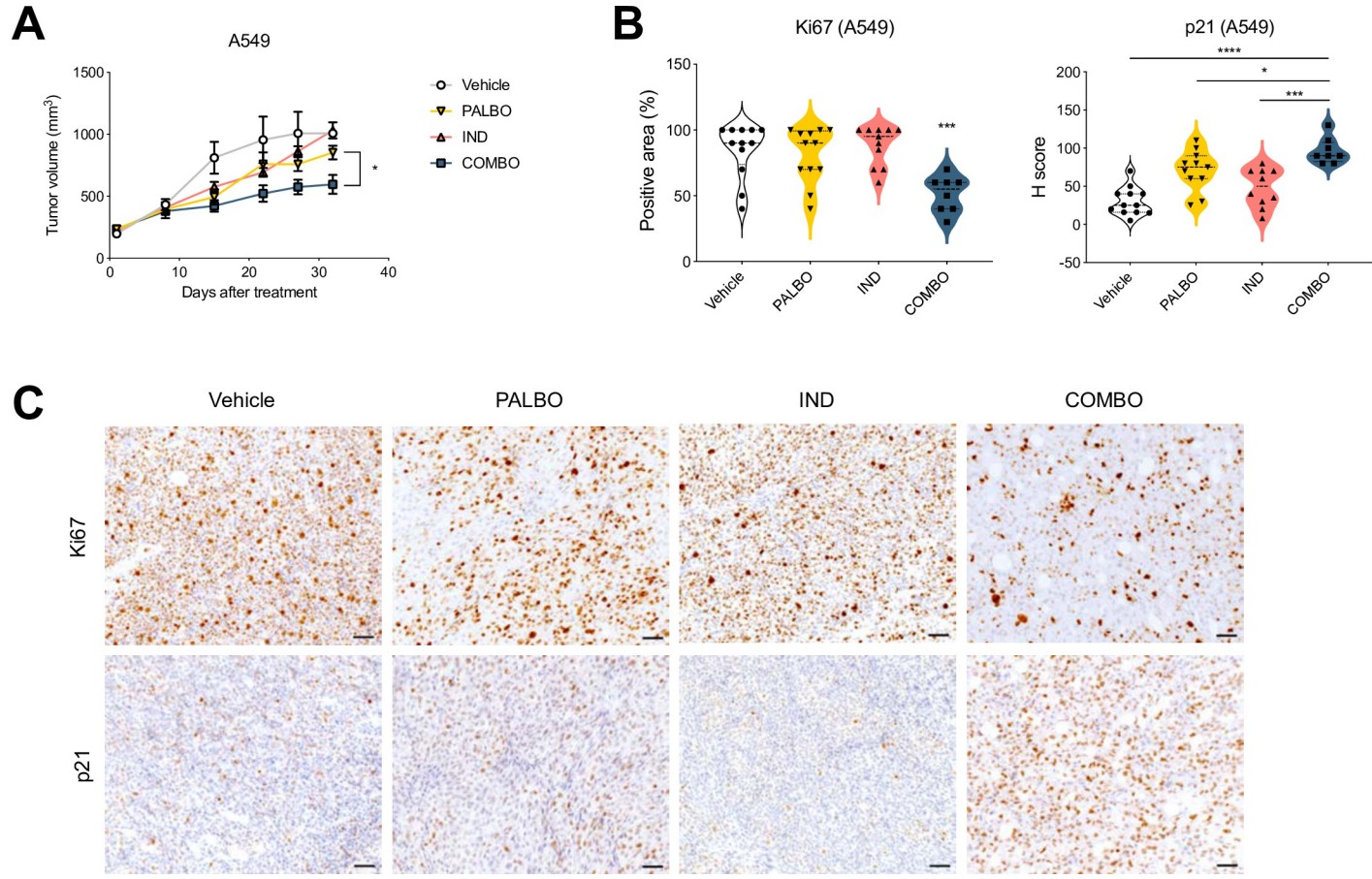

**Fig 4. Combination of indisulam and palbociclib impairs tumor growth *in vivo*. A** Tumor growth of A549 xenografts in NMRI nude mice. Upon tumors reaching 200 mm³ the mice were randomly assigned to treatment with vehicle, indisulam, palbociclib or combination. Palbociclib was administered by oral gavage daily at 100 mg/kg and indisulam by intraperitoneal injection three times per week at 5 mg/kg. Every group consisted of 8–12 mice. Error bars indicate SEM. Tumor volumes of palbociclib and combination treated mice at end point were analysed using unpaired t-test ($^*p<0.05$). **B** Quantification of IHC staining for Ki67 and p21 of A549 tumor xenografts (n = 8–12). Positive area of the tumor was quantified in tumors treated with vehicle, palbociclib, indisulam or combination. One-way ANOVA was performed with Dunnett's post-hoc test ($^*p<0.05$; $^{***}p<0.001$; $^{****}p<0.0001$). For the Ki67 staining, combination is compared to vehicle or single treatments. **C** Representative images from (B) of IHC staining for Ki67 and p21 of A549 tumor xenografts treated with vehicle, palbociclib, indisulam or combination. Images were taken at 20x magnification and the scale bar indicates 50 μm.

overexpressed CDK2 compared to control cells (Fig 5N and S5E Fig in S1 File). Senescence induction was not fully rescued by CDK2 overexpression, which is in line with the observation that the activity and not only abundance of CDK2 determine senescence induction. Furthermore, CDK2 independent effects of indisulam might play a role in senescence induction as well. Taken together, indisulam downregulates *CCNH*, which prevents CDK2 activation and leads to senescence induction when combined with palbociclib (Fig 5O).

## Discussion

Since proliferation of cancer cells depends heavily on the core components of the cell cycle machinery, inhibitors of CDKs could have significant anticancer activity in many tumor types. However, translating the success of CDK4/6 inhibitors from HR+ breast cancer to other cancer types has been hampered by intrinsic resistance [23, 50]. Here we identify a treatment strategy that exploits the synergy between CDK2 loss (i.e., indisulam treatment) and palbociclib treatment in induction of senescence in a diverse panel of cell lines.

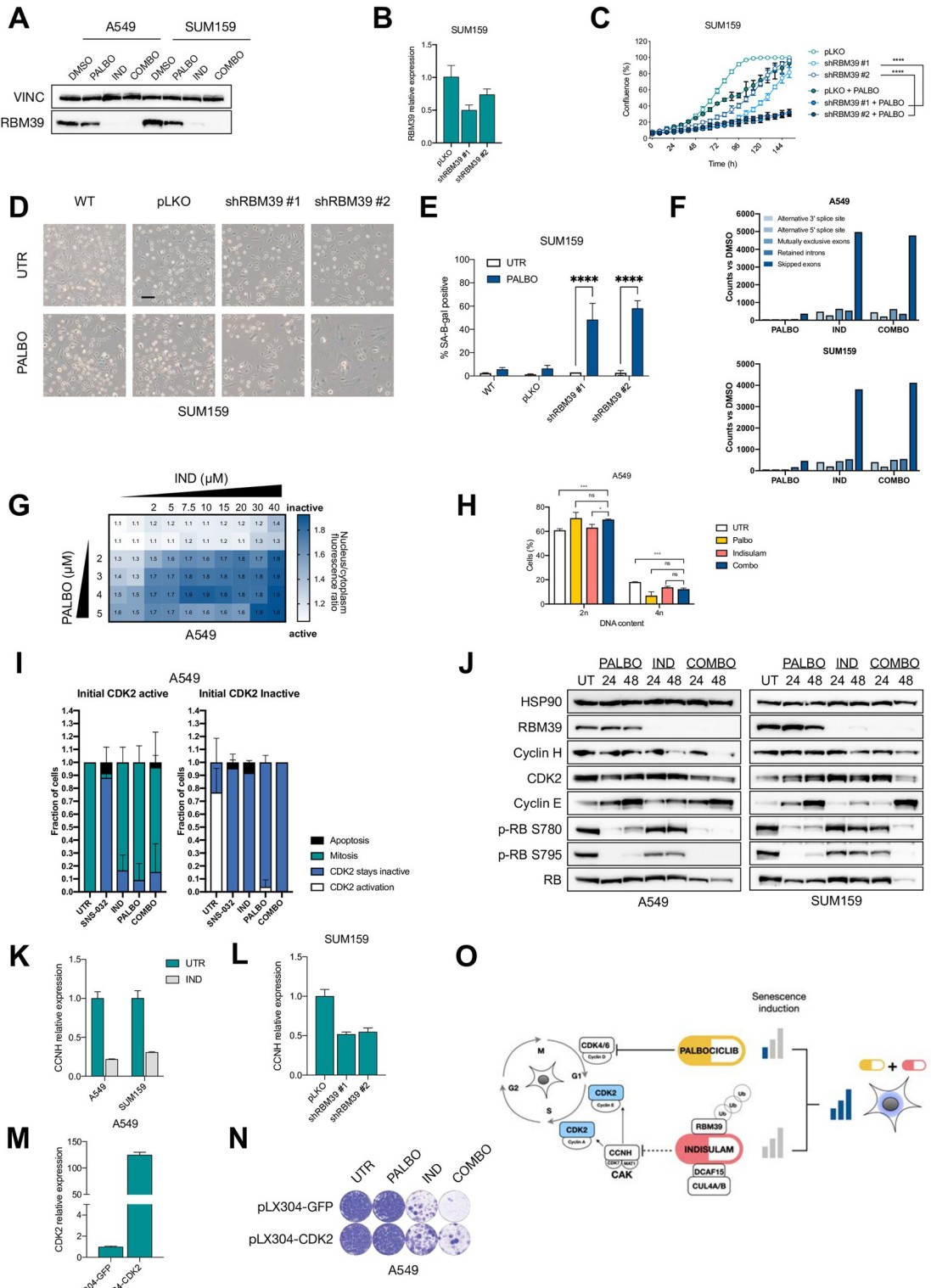

**Fig 5. Indisulam prevents activation of CDK2 leading to cell cycle arrest when combined with palbociclib. A** Western blot analysis of RBM39 in SUM159 and A549 cells treated with palbociclib, indisulam or combination. SUM159 were treated with 0.5 μM of palbociclib and 2 μM indisulam and A549 with 2 μM palbociclib and 0.5 μM indisulam for 12 days. Vinculin was used as a loading control. Representative images of three independent experiments are shown (n = 3). **B-E** RBM39 was knocked-down in SUM159 cells with two independent shRNAs. **B** qPCR analysis of RBM39 normalized to housekeeping gene RPL13 in

SUM159. Mean of three technical replicates is shown and error bars indicate standard deviation. **C** Proliferation assay was performed in RBM39 knock-down and control SUM159 using 0.25 μM of palbociclib. Mean of three technical replicates representative of two independent experiments (n = 2) is shown and error bars indicate standard deviation. The end point confluency of all conditions were analysed using two-way ANOVA with Šidák's post-hoc test (**** $p< 0.0001$). **D** SA-β-gal staining in RBM39 knock-down and control SUM159 cells treated with 0.125 μM of palbociclib for 10 days. Scale bar indicates 100 μm. Representative images of two independent experiments are shown (n = 2). **E** Quantification of SA-β-gal positive cells shown in **D**. SUM159 cells were treated for 10 days with 0.125 μM of palbociclib. Bars represent mean ± SD of triplicates. Data was analysed using two-way ANOVA with Šidák's post-hoc test (**** $p< 0.0001$). **F** Quantification of splicing errors in RNA-sequencing data in A549 and SUM159 cells treated for 16h with 2 μM palbociclib, 3 μM indisulam and the combination, in technical duplicates. Bars represent counts compared to untreated samples. **G** A549 cells expressing a CDK2 reporter DHB-iRFP and H2B-GFP were treated with a matrix of different concentrations of palbociclib (2–5 μM) and indisulam (2–40 μM). After 24h, CDK2 reporter localization and nuclei were imaged by spinning disk microscopy, and the nucleo/cytoplasmic distribution of the CDK2 reporter was analyzed using ImageJ. Matrix shows the average CDK2 reporter ratio of two independent experiments (n = 2). **H** Flow cytometry analysis of A549 cells treated with 5 μM of palbociclib, 40 μM of indisulam or combination for 24h. Mean of three technical replicates representative of two independent experiments (n = 2) is shown and error bars represent standard deviation. Unpaired t-test was performed (*$p<0.05$; **$p<0.01$). **I** A549 cells expressing CDK2 reporter DHB-iRFP and H2B-GFP were treated with 10 μM of SNS-032, 40 μM of indisulam or 5 μM palbociclib and immediately imaged for 24h. CDK2 activity was quantified in cells with initial high CDK2 activity (left plot) and cells with initial low CDK2 activity (right plot). At least 10 cells per condition were quantified and results show the mean of two independent experiments (n = 2). Error bars indicate standard deviation. **J** Western blot analysis of A549 and SUM159 cells treated with 2 μM palbociclib, 5 μM indisulam and the combination for either 24 or 48 hours. HSP90 was used as a loading control. Representative images of three independent experiments are shown (n = 3). **K** qPCR analysis of A549 and SUM159 cells treated with 3 μM of indisulam for 24h. CCNH expression was normalized to the housekeeping gene RPL13. Mean of three technical replicates is shown and error bars indicate standard deviation. **L** qPCR analysis of SUM159 cells harbouring shRBM39 or control cells for CCNH (as in B). Expression was normalised to the housekeeping gene RPL13. Mean of three technical replicates is shown and error bars indicate standard deviation. **M** qPCR analysis of CDK2 overexpressing A549 cells compared to GFP control cells. Expression was normalized to the housekeeping gene RPL13. Mean of three technical replicates is shown and error bars indicate standard deviation. **N** Long term colony formation assay of CDK2 and GFP overexpressing A549 cells treated with 0.5 μM palbociclib, 0.5 μM indisulam and the combination for 10 days. **O** Schematic overview of palbociclib and indisulam senescence induction. Palbociclib inhibits CDK4/6 leading to weak senescence induction and increased dependence on CDK2 for cell cycle progression. Indisulam induces RBM39 degradation through DCAF15 and CUL4A/B leading to splicing errors and downregulation of CCNH. As CCNH is part of CDK2 activating CAK complex, indisulam treatment prevents the activation of CDK2 through CAK. Combination of palbociclib and indisulam therefore induces strong senescent phenotype in cancer cells.

Genetic screens are a powerful tool to identify genetic dependencies in an unbiased manner [51]. Here, we identified *CDK2* loss as an enhancer of the palbociclib effect using genetic screens in three TNBC cell lines. We further validated this interaction in a diverse panel of cell lines, demonstrating that this synergy shows little context dependency. The interaction between *CDK2* loss and CDK4/6 inhibition is not surprising, as CDK2 acts downstream of CDK4/6 in the cell cycle progression. Furthermore, the majority of clinically relevant resistance mechanisms to CDK4/6 inhibition, such as loss of RB and overexpression of *CCNE [20]*, could be circumvented by CDK2 inhibition. As such, *CDK2* depletion was previously shown to re-sensitize both CDK4/6 inhibitor sensitive and resistant cells [22, 24, 43, 52, 53].

Unfortunately, the lack of a specific CDK2 inhibitor has until now prevented exploiting the synergy between CDK2 inhibition and CDK4/6 inhibition. In spite of this, multiple targeting strategies were previously described to increase sensitivity to CDK4/6 inhibition, such as using a multi CDK inhibitor roscovitine [24], knock-in of analog-sensitive CDK2 [22] and use of triple CDK2/4/6 inhibitor PF-06873600 [43, 44]. Unfortunately, the use of roscovitine in the clinic is hindered by off-target effects and toxicity, and analog-sensitive CDK2 lacks translational potential. Even though PF-06873600 is in clinical development (phase 1 clinical trial ongoing: NCT03519178), the compound also inhibits CDK1, which might lead to toxicity as seen with other multi-CDK inhibitors [43]. Interestingly, comparing palbociclib to two other CDK4/6 inhibitors abemaciclib and ribociclib shows that abemaciclib is more effective compared to palbociclib and ribociclib, possibly due to the notion that it also targets CDK2 [45]. However, both intrinsic and acquired resistance are still an issue, indicating a need for a different therapeutic strategy. We propose using the indirect CDK2 inhibitor indisulam, which has

previously shown a favorable toxicity profile in the clinic. We demonstrate that combination of palbociclib and indisulam induces senescence in a diverse cell line panel, which points to a broad applicability across tumor types. Additionally, our preliminary data indicate that the combination can be effective in vivo.

Even though indisulam was described as an indirect CDK2 inhibitor, later studies revealed that it targeted a splicing factor—RBM39—for degradation. Interestingly, we observed an effect of indisulam on CDK2 activity only in cells where CDK2 is initially inactive, indicating that indisulam prevents CDK2 activation. Regulation of CDK2 activity is based on its interaction with cyclins, removal of inhibitory phosphorylation by CDC25 and activating phosphorylation by the CDK Activating Kinase (CAK) complex. We observed that cyclin H, which is a part of the CAK complex, is downregulated upon indisulam treatment in an RBM39 dependent manner. As we observed no splicing errors in *CCNH* transcripts or other CDK2 regulators, it is still unclear how reduction of RBM39 leads to *CCNH* downregulation. As *CCNH* is an essential gene, loss of function genetic experiments are technically challenging to perform. Additionally, other CDKs might be involved in senescence induction through indisulam since CAK regulates CDK1, 4 and 6 in addition to CDK2. Furthermore, as activity and not amount of CDK2 seems to play a role in indisulam sensitivity, overexpression of CDK2 is at best expected to partially rescue the senescence induction, which is indeed what we observed here. Lastly, we observed accumulation of thousands of splicing errors in cells treated with indisulam, which might sensitize the cells to palbociclib in CDK2 independent manner and further explain the partial rescue of CDK2 overexpression.

Combining palbociclib with indisulam might be a potential treatment strategy for cell types that are intrinsically resistant to palbociclib. In addition, acquired resistance to palbociclib has been shown to be reversed by depleting CDK2. For example, loss of RB leads to resistance to palbociclib and the combination of palbociclib with a MEK inhibition [26]. However, RB deficient cells are still sensitive to knockdown of CDK2 [53]. It is therefore likely that the combination of palbociclib and indisulam would still be effective in palbociclib resistant cells, although the mechanism of senescence induction in RB deficient cells is not yet well understood. Furthermore, recent reports on Cyclin D regulation described *AMBRA1* loss as a resistance mechanism to CDK4/6 inhibition [54], but as those cells still depend on CDK2 activity, combination with indisulam could still be effective. Senescence induction is increasingly in focus as a potential cancer therapeutic strategy, supported by the findings that senolytic compounds can be effective in eradicating senescent cancer cells [6]. Here, we have shown that senescence induction by palbociclib and indisulam sensitizes the cells to the established senolytic drug navitoclax. Finally, senescent cells attract immune cells, and together with the notion that indisulam induced splicing errors could lead to generation of neoantigens, the combination of senescence induction and immunotherapy might be a potential future treatment strategy [26, 55].

## 2. Supplemental table

List of senescence gene signatures and gene sets of genes unregulated in senescence used to assess the senescence state through RNA sequencing

## 3. Supplemental images

Uncropped Western blot images shown in figures and supplemental figures

## Materials availability

Plasmid generated in this study is available from the corresponding authors upon request.

## Methods

### Cell lines

TNBC cell line CAL-51 was grown in DMEM (Gibco) supplemented with 20% fetal bovine serum (FBS, Serana), 1% penicillin-streptomycin (P/S, Gibco) and 2mM L-glutamine (Gibco). TNBC cell line CAL-120 was grown in DMEM supplemented with 10% FBS, 1% P/S and 2 mM L-glutamine. TNBC cell line HCC1806, lung cancer cell lines A549 and H2122, colon cancer cell lines DLD-1 and RKO were grown in RPMI (Gibco) supplemented with 10% FBS, 1% P/S and 2 mM L-glutamine. TNBC cell line SUM159 was grown in DMEM/F12 (Gibco) supplemented with 10% FBS, 1% P/S, 5 μg/ml insulin (Sigma-Aldrich) and 1μg/ml hydrocortisone (Sigma-Aldrich).

HCC1806, A549, RKO, H2122 and DLD-1 were purchased from ATCC. SUM159 was a gift from Metello Innocenti (NKI, Amsterdam). CAL-51 and CAL-120 were obtained from DSMZ. All cell lines were regularly tested for mycoplasma contamination using a PCR assay and STR profiled (Eurofins).

### Compounds and antibodies

Palbociclib, indisulam and ABT-263 were purchased from MedKoo (Cat:#123215, #201540 and #201970). PF-06873600 was purchased from Selleck chem. Antibodies against CDK2 and tubulin were purchased from Abcam. Antibodies against vinculin and p-H2AX S139 were purchased from Sigma Aldrich. Antibodies against p-RB S780, p-RB S795, RB (9309-4H1), Lamin B1, Cyclin H and Cyclin E were purchased from Cell Signalling Technology. Antibodies against HSP90 and p21 were purchased from Santa Cruz Biotechnology. Antibody against RBM39 was purchased from Atlas Antibodies. Antibody against p16 was purchased from Proteintech.

### Kinome dropout shRNA screens

TNBC cell lines CAL-51, CAL-120 and HCC1806 were screened using a kinome shRNA library targeting 518 human kinases and 17 kinase-related genes. The kinome library was assembled from the RNAi Consortium (TRCHs 1.5 and 2.0) shRNA collection and included 243 hairpins targeting essential and 272 targeting non-essential genes. Upon generating lentiviral shRNA vectors, CAL-51, CAL-120 and HCC1806 cells were infected using a low infection efficiency of <30%, reference sample t = 0 was collected and cells were then cultured in the presence or absence of palbociclib (0.4 μM for CAL-51, 1 μM for CAL-120 and 0.2 μM for HCC1806), while maintaining 1000x coverage of the library. shRNA sequences were then recovered by PCR from genomic DNA, and the abundance was quantified by deep sequencing. The analysis was performed using DESeq [56]. Hit selection was done in two steps: initially the hits were selected based on the comparison of treated to untreated arm with the criteria of at least two shRNAs per gene with log2 fold change <-1, FDR <0.1 and baseMeanA > 100 and no hit shRNAs in the opposite direction. To exclude the shRNAs that are increased in the untreated condition, instead of decreased in treated compared to reference t = 0 sample, we performed an additional selection step in which sgRNA should have log2 fold change < -1 in treated condition compared to reference t = 0 condition. Hits that overlapped between the three cell lines were prioritized for validation.

### Plasmids

The lentiviral shRNA vectors were selected from the arrayed TRC human genome-wide shRNA collection in pLKO backbone. shRNA targeting CDK2 #1: CTATGCCTGATTAC AAGCCAA, shRNA targeting CDK2 #2: GCCCTCTGAACTTGCCTTAAA, shRNA targeting

RBM39 #1: `GCCGTGAAAGAAAGCGAAGTA`, shRNA targeting RBM39 #2: `GCTGGACCTAT GAGGCTTTAT`. Single gRNAs were cloned into LentiCRISPR 2.1 plasmid [57] by BsmBI (New England BioLabs) digestion and Gibson Assembly (New England BioLabs); control sgRNA: `ACGGAGGCTAAGCGTCGCAA`, sgRNA targeting CDK2 #1: `GTTCGTACTTACACCCATGG`, sgRNA targeting CDK2 #2: `CATGGGTGTAAGTACGAACG`. For overexpression experiments pLX304-Blast-V5 were used, with either GFP or CDK2 (ID ccsbBroad304_00276). Additionally, pCMV-GFP (Addgene #11153) and pCMV-CDK2 [58] were used to transiently transfect the target cells.

## Lentiviral transduction

Second generation lentivirus packaging system consisting of psPAX2 (Addgene #12260), pMD2.G (Addgene #12259) and pCMV-GFP as transfection control (Addgene #11153) was used to produce lentivirus particles. After transient transfection in HEK293T cells using polyetylenamine (PEI), lentiviral supernatant was filtered and used to infect target cells using 8 mg/ml polybrene. After infection, the cells were selected with 2 mg/ml puromycin or 10 mg/ml blasticidin. After 48-72h or until non-transduced control cells were dead, the selection was complete.

## RNA sequencing and GSEA

Total RNA was extracted with RNeasy mini kit (Qiagen, cat# 74106) including a column DNase digestion (Qiagen, cat#79254), according to the manufacturer's instructions. Quality and quantity of total RNA was assessed by the 2100 Bioanalyzer using a Nano chip (Agilent, Santa Clara, CA). Total RNA samples having RIN>8 were subjected to library generation. Strand-specific libraries were generated using the TruSeq Stranded mRNA samples preparation kit (illumine Inc., San Diego, RS-122-2101/2) according to manufacturer's instructions (Illumina, part #15031047 Rev.E). Briefly, polyadenylated RNA from intact total RNA was purified using oligo-dT beads. Following purification, the RNA was fragmented, random primed and reverse transcribed using SuperScript II Reverse Transcriptase (Invitrogen, part # 18064–014) with the addition of Actinomycin D. Second strand synthesis was performed using Polymerase I and RNaseH with replacement of dTTP for dUTP. The generated cDNA fragments were 3' end adenylated and ligated to Illumina Paired-end sequencing adapters and subsequently amplified by 12 cycles of PCR. The libraries were analyzed on a 2100 Bioanalyzer using a 7500 chip (Agilent, Santa Clara, CA), diluted and pooled equimolar into a multiplex sequencing pool. The libraries were sequenced with single-end 65bp reads on a HiSeq 2500 using V4 chemistry (Illumina inc., San Diego).

For the analysis, reads were first aligned to a reference genome (hg38) and the datasets were normalized for sequence depth using a relative total size factor. We then performed gene set enrichment analysis (GSEA) using GSEA software [59] with log2FoldChange ranked list as an input. The GSEA preranked tool was used to run the analysis. We used two senescence gene signatures [2, 39] as well as gene sets of genes upregulated in senescence from [40, 41] to assess enrichment of senescence- associated genes (S1 Table).

In genetic experiments we compared sgCDK2 cells treated with palbociclib with untreated cells and for pharmacological experiments cells treated with palbociclib and indisulam to untreated cells or single treatments. When using PF-0687360 we compared cells treated with the compound to untreated cells. All experiments were performed in duplicates. The *P*-value estimates the statistical significance of the enrichment score and is shown in the figure, unless $P < 0.001$.

## Splicing error quantification

The RNA was isolated and libraries were prepared as described above. The libraries were sequenced with 75bp paired-end reads on a NextSeq550 using the High Output Kit v2.5, 150 Cycles (Illumina Inc., San Diego). For the analysis, sequences were demultiplexed and adapter sequences were trimmed from using SeqPurge [60]. Trimmed reads were aligned to GRCh38 using Hisat2 [61] using the prebuilt genome_snp_tran reference. Splice event detection was performed using rMats version 4.0.2 by comparing the replicates of the treated groups to the replicates of the untreated group [62]. rMats events in the different categories were considered significant when the following thresholds were met: having a minimum of 10 reads, an FDR less than 10% and an inclusion-level-difference greater than 10%, as described earlier [63].

## Kinase enrichment analysis

RNa-seq data generated from the splicing experiment was filtered for adjusted p-value <0.05 and analysed using the Enrichr software [64] using kinase enrichment analysis [49].

## CDK2 activity experiments

CDK2 reporter DHB-iRFP was modified from DHB-Venus (Addgene #136461) [46]. Venus was replaced for iRFP713 through Gibson Assembly. In brief, iRFP713 was amplified adding sequence homology and assembled into the BamHI and HpaI sites of the original CDK2 reporter plasmid. The following primers were used to amplify iRFP713; F:`ACCGATAATCAA‑GAAACTGGATCCGGGGCCCAGGGCAGCGGCATGGCGGAAGGCTCCGTC` R: `GTTGATTATCGATAAGCTTGATCCCTCGATGCGGCCGCTTACTCTTCCATCACGCCGATC`.

A549 cells were stably transduced with a lentiviral vector containing H2B-GFP (Addgene #25999) and a lentiviral vector containing DHB-iRFP, and subsequently GFP/iRFP double positive cells were isolated by FACS. In order to determine CDK2 activity following 24h drug treatment, cells were seeded in 96 well plate, treated with indisulam and palbociclib for 24h and then imaged on a spinning disc microscope using Andor 505 Dragonfly system equipped with 20x 0.75 NA objective and Zyla 4.2+, sCMOS camera. CDK2 activity was determined by calculating the nucleo/cytoplasmic ratio of the CDK2 reporter, using a custom macro for Ima-geJ, as described before [65]. For real-time analysis of CDK2 activity immediately following drug treatment, cells were seeded in chambered covered slides (LabtekII), and subsequently imaged for 25h. Inhibitors were added following the first round of image acquisition. Imaging was performed using a Deltavision Elite (Applied Precision) that was maintained at 5% $CO_2$ and 37°C, equipped with a 20x 0.75 NA lens (Olympus) and cooled Hamamatsu ORCA R2 Black and White CCD-camera.

## Flow cytometry

A549 cells were treated for 24h with indicated doses of palbociclib and indisulam. Cells were then harvested, fixed in ice cold 70% ethanol and stained with DAPI 8 μg/ml in PBS. Samples were acquired with LSRFortessa (BD Biosciences) and analysis was performed with FlowJo10 software. Single cells were gated via DAPI-A and DAPI-H signals and DNA content was gated based on DAPI-A histogram profile.

## Western blot

Cells were lysed using RIPA buffer and protein was extracted and quantified using BCA assay (Pierce). Loading buffer and reducing agent (both Thermo Fisher) were added to the samples, which were then boiled for 5 min at 95°C. The samples were resolved on a 5–15% Bis-Tris gel

(Thermo Fisher) followed by blotting. Membranes were incubated with primary antibodies diluted to 1:1000 in 5% BSA. Secondary antibodies were used at 1:10000 dilution and were purchased from Biorad. Signal was visualized by the ECL solution (Biorad) using the Chemi-Doc Imaging system (Biorad). To quantify, the intensity of each band was measured using ImageJ and normalized to the loading control and untreated condition. All uncropped Western blot images are provided as supplemental material (Supplemental Images 1).

### Kinase inhibition assay

Inhibition of CDK/cyclin complexes by indisulam was measured by Z'LYTE—SelectScreen Kinase Profiling Services (ThermoFisher). Briefly, CDK/cyclin complexes were incubated with indisulam, substrates and ATP. The kinase activity of the CDK/cyclin complex was measured as ATP consumption, as described in detail previously [45].

### Quantitative reverse transcription PCR

RNA was extracted using Isolate II Mini kit (Bioline), following manufacturer's instructions. cDNA was generated using SensiFast cDNA synthesis kit (Bioline) following manufacturer's instructions. For qPCR reaction 1 μg of cDNA was used with SensiFast Sybr Lo-Rox mix (Bioline) and respective primer pair. All reactions were performed in triplicates and the results were analyzed using the deltadelta Ct method. The sequences of primers used are as follows:

*RPL13* forward `GGCCCAGCAGTACCTGTTTA`, *RPL13* reverse `AGATGGCGGAGGTGCAG`, *RBM39* forward `GTCGATGTTAGCTCAGTGCCTC`, *RBM39* reverse `ACGAAGCATATCTTCAG TTATG`, *CCNH* forward `TGTTCGGTGTTTAAGCCAGCA`, *CCNH* reverse `TCCTGGGGTGATA TTCCATTACT`.

### Senescence associated B-galactosidase staining

Cells were stained using the Senescence Cells Histochemical Staining kit (CS0030) from Sigma Aldrich according to the manufacturer's instructions. Stained cells were imaged at 100x magnification and at least 3 pictures per condition were taken. The staining was quantified by counting at least 100 cells from 3 independent images. To evaluate the increase in SA-β-gal positive cells in combination treatment compared to individual drug treatments we performed one-way ANOVA comparing each treatment group to the combination.

### Colony formation assay and proliferation assay

Cells were plated in 6-well plates with densities between 10–40000 cells per well, depending on the cell line. Medium and drugs were refreshed every 3–4 days. After 10 days the cells were fixed with 4% formaldehyde (Millipore) in PBS, stained with 2% Crystal Violet (Sigma) in water and scanned. For proliferation assays, cells were plated in 96-well plates with densities between 500–2000 cells per well, depending on cell line. Plates were incubated at 37˚C and images were taken every 4 hours using the IncuCyte ® live cell imaging system. Medium and drugs were refreshed every 3–4 days. Confluency was calculated to generate growth curves.

### In vivo experiments

All animal experiments were approved by the Animal Ethics Committee of the Netherlands Cancer Institute and were performed in accordance with institutional, national and European guidelines for Animal Care and Use. CDK2 KO clones used in vivo were generated through transient transfection of a plasmid containing Cas9 and gRNA sequences, followed by the brief puromycin selection and characterisation of the clones. For the genetic experiments one

million of CAL-51 CDK2 KO single cell clone or control cells in PBS were mixed 1:1 with matrigel and injected orthotopically in the mammary fat pad of 8 weeks female NMRI nude mice (JAX labs), 5–6 mice per group. Furthermore, one million of A549 CDK2 KO single cell clones or control cells in PBS were mixed 1:1 with matrigel and injected subcutaneously in the right flank of NMRI nude mice, 5–6 mice per group. Tumor volume was monitored twice a week and tumor volume was calculated based on the calliper measurements following modified ellipsoidal formula (tumor volume = ½[length x width2]). For the intervention experiment, one million of A549 cells in PBS were mixed 1:1 with matrigel and injected subcutaneously in the right flank of NMRI nude mice. Upon reaching 200 mm3, mice were randomized to four treatment groups of 8–12 mice per group: vehicle, indisulam, palbociclib or combination. Palbociclib (dissolved in 50 mM sodium lactate) was administered by oral gavage daily at 100 mg/kg and indisulam (dissolved in 3.5% DMSO, 6.5% Tween 80, 90% saline) by intraperitoneal injection three times per week at 5 mg/kg. Four mice were excluded from the study due to complications of daily intraperitoneal injections (peritonitis). Anesthesia was administered in a form of isoflurane (3% induction, 2%maintenece) and analgesia in form of carprofen. The mice were sacrificed using CO2. The following humane end points were applied according to the Code of Practice Animal Experiments in cancer research to alleviate suffering:

- Waiting for spontaneous death is NOT allowed;

- Animal loses more than 15% of body weight within 2 days;

- Animal has lost more than 20% of body weight since start of experiment;

- Animal has circulation or breathing problems;

- Animal shows aberrant behaviour/movement;

- The tumour causes clinical symptoms (as a result of location, invasive growth or ulceration);

- The tumour has reached the size of more than 10% of the normal bodyweight or has a diameter of 15 mm (is approx. 2 cm3).

## Immunohistochemistry

Tumors were collected and fixed in EAF fixative (ethanol/acetic acid/formaldehyde/saline at 40:5:10:45 v/v) and embedded in paraffin. For immunohistochemistry, 4 μm-thick sections were made on which antibodies against Ki67 and p21 were applied. The sections were reviewed with a Zeiss Axioskop2 Plus microscope (Carl Zeiss Microscopy) and images were captured with a Zeiss AxioCam HRc digital camera and processed with AxioVision 4 software (both from Carl Zeiss Vision). Histological samples were analyzed by an experienced pathologist. Scoring was performed by quantifying positive area for Ki67 and H-score for p21.

## PK/PD experiment for PD and IN

Blood was collected either from the tail vein or by cardiac puncture at different time points indicated in S4C and S4D Fig in S1 File. Samples were collected on ice using tubes with potassium EDTA as anticoagulant. After cooling, tubes were centrifuged for 10 min 5000xg 4˚C to separate the plasma fraction, which was then transferred into clean vials and stored at −20˚C until analysis. Sample pre-treatment was accomplished by mixing 5 μL (plasma) with 60 μL of formic acid in acetonitrile (1 + 99) containing the internal standard. After centrifugation, the clear supernatant was diluted 1 + 8 with water and 5 μL was injected into the LC-MS/MS

system. The samples were assayed twice by liquid chromatography triple quadrupole mass spectrometry (LC-MS/MS) using an API4000 detector (Sciex). Indisulam is detected in negative ionization mode (MRM: 384.2/171.9) and palbociclib in positive ionization mode (MRM: 448.5/320.0). In both cases, LC separation was achieved using a Zorbax Extend C18 column (100 x 2.0 mm: ID). Mobile phase A and B comprised 0.1% formic acid in water and methanol, respectively. The flow rate was 0.4 ml/min and a linear gradient from 20%B to 95%B in 2.5 min, followed by 95%B for 2 min, followed by re-equilibration at 20%B for 10 min was used for elution.

## Statistical analysis

Unpaired t-test and ANOVA were performed with Graphpad Prism (v8.4.3).

## Supporting information

**S1 File.**
(PDF)

**S2 File. Uncropped Western blot images shown in figures and supplemental figures.**
(PDF)

**S1 Table. List of senescence gene signatures and gene sets of genes unregulated in senescence used to assess the senescence state through RNA sequencing.**
(XLSX)

## Acknowledgments

We would like to thank Artur Burylo, Natalie Proost, Jordi van Iersel and Marcio Sousa for help with experiments and Katrien Berns for critical discussion on screening techniques. We also thank the NKI Bioimaging facility, Genomics core facility and pre-Clinical Intervention Unit for their technical support.

## Author Contributions

**Conceptualization:** Ziva Pogacar, Jackie L. Johnson, Rene Bernards, Rodrigo Leite de Oliveira.

**Data curation:** Cor Lieftink, Arno Velds.

**Formal analysis:** Giulia De Conti, Fleur Jochems, Cor Lieftink, Arno Velds, Leyma Wardak, Kelvin Groot, Ji-Ying Song.

**Funding acquisition:** Rene H. Medema, Rene Bernards.

**Investigation:** Ziva Pogacar, Jackie L. Johnson, Lenno Krenning, Giulia De Conti, Fleur Jochems, Leyma Wardak, Kelvin Groot, Arnout Schepers, Liqin Wang, Ji-Ying Song, Rodrigo Leite de Oliveira.

**Methodology:** Lenno Krenning.

**Software:** Cor Lieftink, Arno Velds, Kelvin Groot.

**Supervision:** Marieke van de Ven, Olaf van Tellingen, Rene H. Medema, Roderick L. Beijersbergen, Rene Bernards, Rodrigo Leite de Oliveira.

**Validation:** Ziva Pogacar, Jackie L. Johnson, Lenno Krenning.

**Visualization:** Ziva Pogacar, Lenno Krenning, Giulia De Conti, Fleur Jochems, Cor Lieftink, Arno Velds, Leyma Wardak, Kelvin Groot, Rodrigo Leite de Oliveira.

**Writing – original draft:** Ziva Pogacar, Rene Bernards, Rodrigo Leite de Oliveira.

**Writing – review & editing:** Ziva Pogacar, Arnout Schepers, Liqin Wang, Roderick L. Beijersbergen, Rodrigo Leite de Oliveira.

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
