## [Decision Letter · Decision Letter 0]

23 Dec 2021

PONE-D-21-35335Indisulam synergizes with palbociclib to induce senescence through inhibition of CDK2 kinase activityPLOS ONE

Dear Dr. Leite de Oliveira,

Thank you for submitting your manuscript to PLOS ONE. After careful consideration, we feel that it has merit but does not fully meet PLOS ONE’s publication criteria as it currently stands. Therefore, we invite you to submit a revised version of the manuscript that addresses the points raised during the review process.

We look forward to receiving your revised manuscript.

Kind regards,

Yuan-Soon Ho

Academic Editor

PLOS ONE

Journal Requirements:

We would like to thank Artur Burylo, Natalie Proost, Jordi van Iersel and Marcio Sousa for help with experiments and Katrien Berns for critical discussion on screening techniques. We also thank the NKI Bioimaging facility, Genomics core facility and pre-Clinical Intervention Unit for their technical support. This work was supported by grants from the European Research Council (ERC) to R.B. and the Dutch Cancer Society through the Oncode Institute.

EC | H2020 | H2020 Priority Excellent Science | H2020 European Research Council (ERC):Ziva Pogacar,Jackie L. Johnson,Giulia De Conti,Cor Lieftink,Leyma Wardak,Fleur Jochems,Kelvin Groot,Arnout Schepers,Liqin Wang,Roderick Beijersbergen,Rene R Bernards,Rodrigo Leite de Oliveira 787925

Other authors worked without finding.

Rene Bernards is the founder of the company Oncosence (https://www.oncosence.com), which aims to develop senescence-inducing and senolytic compounds to treat cancer.

7. PLOS ONE now requires that authors provide the original uncropped and unadjusted images underlying all blot or gel results reported in a submission’s figures or Supporting Information files. This policy and the journal’s other requirements for blot/gel reporting and figure preparation are described in detail at https://journals.plos.org/plosone/s/figures#loc-blot-and-gel-reporting-requirements and https://journals.plos.org/plosone/s/figures#loc-preparing-figures-from-image-files. When you submit your revised manuscript, please ensure that your figures adhere fully to these guidelines and provide the original underlying images for all blot or gel data reported in your submission. See the following link for instructions on providing the original image data: https://journals.plos.org/plosone/s/figures#loc-original-images-for-blots-and-gels. 

Reviewers' comments:

Reviewer's Responses to Questions

**Comments to the Author**

1. Is the manuscript technically sound, and do the data support the conclusions?

Reviewer #1: Yes

Reviewer #2: Partly

2. Has the statistical analysis been performed appropriately and rigorously? 

Reviewer #1: Yes

Reviewer #2: No

3. Have the authors made all data underlying the findings in their manuscript fully available?

Reviewer #1: Yes

Reviewer #2: Yes

4. Is the manuscript presented in an intelligible fashion and written in standard English?

Reviewer #1: Yes

Reviewer #2: Yes

5. Review Comments to the Author

Reviewer #1: Authors tried to evaluate the induction of senescence in cancer cells for a new therapeutic strategy. They found that blocking CDK2 by indisulam, a novel sulfonamide anticancer agent (N-(3-chloro-7-indolyl)-1,4-benzenedisulfonamide, E7070), would enhance senescence induction by palbociclib, a CDK4/6 inhibitor approved for treatment of metastatic breast cancer, in triple negative breast cancer (TNBC) cells, various cell lines and lung cancer xenografts. Thus, they concluded that combined treatment with palbociclib and indisulam induced a senescence program and sensitized cells to senolytic therapy, illustrating that inhibition of CDK2 through indisulam treatment can enhance senescence induction by CDK4/6 inhibition. Whole manuscript clearly shows the good rationale, methods, results and figure illustrations. However, there is no space between word and citation parenthesis in the text, which should be correctly reformatted.

Reviewer #2: Major concerns:

1. Previous report demonstrated that indisulam (E7070) inhibited the phosphorylation of pRb, decreased expressions of cyclin A, B1, CDK2, and CDC2 proteins, and suppressed CDK2 catalytic activity with the induction of p53 and p21 proteins in A549 cells but not in A549/ER cells (Invest New Drugs 2001;19:219-27). Therefore, the combined therapeutic benefit of indisulam and palbociclib still did not prove due to the inhibitory effect of CDK2 by indisulam, the off-target effect of indisulam may also play a role to contribute to this combined benefit.

2. Looking at the research, there is no direct evidence to prove that indisulam synergizes with palbociclib to induce senescence through inhibition of CDK2 kinase activity (Title). First, the author did not present evidence that indisulam inhibits CDK2 kinase activity. Secondly, the author has not proved that indisulam induces senescence by inhibiting CDK2 kinase activity.

3. There is not enough evidence to prove that indisulam is a CDK2 inhibitor in the present study. In fact, the results show that palbociclib alone has a more pronounced inhibitory effect on CDK2 (Figure 3E). The relevant issues need to be clarified.

4. Authors indicated that treating cells with indisulam led to downregulation of cyclin H, which prevented CDK2 activation. However, indisulam did not reduce cyclin H levels till 48 hours of treatment (Figure 5I), but CDK2 activity tests are only analyzed within 24 hours. Such experimental design and results cannot explain that treating cells with indisulam led to downregulation of cyclin H, which prevented CDK2 activation.

5. Authors mentioned that combined treatment with palbociclib and indisulam induced a senescence program and sensitized cells to senolytic therapy. According to the result, this event is only found in A549 cells, but not SUM159 cells (Figure 3D). The relevant issues need to be clarified.

6. There is no evidence to support that COMBO therapy induces tumor senescence in vivo and resensitizes tumors to senolytic therapy, and relevant markers need to be checked.

6. PLOS authors have the option to publish the peer review history of their article (what does this mean?). If published, this will include your full peer review and any attached files.

Reviewer #1: No

Reviewer #2: No

---

## [Author Response · Author response to Decision Letter 0]

24 Feb 2022

The editorial comments are addressed below. 

1. Funding

We have removed the funding information from the acknowledgements section. We made sure that information on funding is available in the Funding statement, below: 

EC | H2020 | H2020 Priority Excellent Science | H2020 European Research Council (ERC):Ziva Pogacar,Jackie L. Johnson,Giulia De Conti,Cor Lieftink,Leyma Wardak,Fleur Jochems,Kelvin Groot,Arnout Schepers,Liqin Wang,Roderick Beijersbergen,Rene R Bernards,Rodrigo Leite de Oliveira 787925

Other authors worked without finding.

2. Competing interests

We have updated the Competing interests section as follows:

R.B is the founder of the company Oncosence (https://www.oncosence.com), which aims to develop senescence-inducing and senolytic compounds to treat cancer. This does not alter our adherence to PLOS ONE policies on sharing data and materials.

3. Repository

We have submitted our raw sequencing data to the repository. We will provide the accession number as soon as it becomes available. 

4. Uncropped images

We have included a supplementary file S1_raw_images where we collected all the raw uncropped images of blots. 

5. Data changes

We were not able to retrieve original images of SA-Bgal experiment in supplemental figure 3C of CAL-51 for quantification. We have therefore included a new experiment and quantification for CAL-51 (Supplemental figure 3C,D).

Similarly, we were not able to locate the original blot of pRB S780 for A549 in figure 3F. We have thus included new data for tubulin and pRB S780 in figure 3F under the dotted line. 

--

We would like to thank the reviewers for their detailed reading of the manuscript and the insightful comments. We believe that the revised manuscript adequately addresses the reviewers’ questions. 

1. Is the manuscript technically sound, and do the data support the conclusions?

Reviewer #1: Yes

Reviewer #2: Partly

2. Has the statistical analysis been performed appropriately and rigorously?

Reviewer #1: Yes

Reviewer #2: No

In order to improve this, we included quantification and statistical testing on SA-β-gal stainings (Figure 1G, 2E and J, 3D, 5E, S1F, S3D and G). Furthermore, we performed statistical testing on all proliferation experiments (Figure 1E, 2C and H, 3B, 5C, S1D, S3B) and growth curves of the in vivo experiment (Figure 4A). 

3. Have the authors made all data underlying the findings in their manuscript fully available?

Reviewer #1: Yes

Reviewer #2: Yes

4. Is the manuscript presented in an intelligible fashion and written in standard English?

Reviewer #1: Yes

Reviewer #2: Yes

5. Review Comments to the Author

Reviewer #1: Authors tried to evaluate the induction of senescence in cancer cells for a new therapeutic strategy. They found that blocking CDK2 by indisulam, a novel sulfonamide anticancer agent (N-(3-chloro-7-indolyl)-1,4-benzenedisulfonamide, E7070), would enhance senescence induction by palbociclib, a CDK4/6 inhibitor approved for treatment of metastatic breast cancer, in triple negative breast cancer (TNBC) cells, various cell lines and lung cancer xenografts. Thus, they concluded that combined treatment with palbociclib and indisulam induced a senescence program and sensitized cells to senolytic therapy, illustrating that inhibition of CDK2 through indisulam treatment can enhance senescence induction by CDK4/6 inhibition. Whole manuscript clearly shows the good rationale, methods, results and figure illustrations. However, there is no space between word and citation parenthesis in the text, which should be correctly reformatted.

We were happy to hear the reviewer agrees that the rationale and methods of our findings are relevant to the field. Furthermore, we have adapted the formatting of citations as suggested. 

Reviewer #2: Major concerns:

1. Previous report demonstrated that indisulam (E7070) inhibited the phosphorylation of pRb, decreased expressions of cyclin A, B1, CDK2, and CDC2 proteins, and suppressed CDK2 catalytic activity with the induction of p53 and p21 proteins in A549 cells but not in A549/ER cells (Invest New Drugs 2001;19:219-27). Therefore, the combined therapeutic benefit of indisulam and palbociclib still did not prove due to the inhibitory effect of CDK2 by indisulam, the off-target effect of indisulam may also play a role to contribute to this combined benefit.

We agree with the reviewer that indisulam likely has other effects in addition to CDK2 inhibition. We propose that indisulam is an indirect CDK2 inhibitor, as overexpression of CDK2 leads to partial rescue of senescence induction by palbociclib and indisulam (figure 5N).Moreover, we included a comment about CDK2 independent effects of indisulam in the discussion (line 268-269). 

2. Looking at the research, there is no direct evidence to prove that indisulam synergizes with palbociclib to induce senescence through inhibition of CDK2 kinase activity (Title). First, the author did not present evidence that indisulam inhibits CDK2 kinase activity. Secondly, the author has not proved that indisulam induces senescence by inhibiting CDK2 kinase activity.

We studied the effects of indisulam treatment on CDK2 activity using a CDK2 reporter in figure 5G and I. We observed that treatment with indisulam prevented CDK2 activation in cells where CDK2 was initially inactive. This suggests that indisulam indirectly prevents CDK2 activity. Secondly, overexpression of CDK2 rescues the senescence induction of cells treated with palbociclib and indisulam as shown in figure 5N. We believe that these experiments support our conclusion that senescence induction by palbociclib and indisulam is mediated by CDK2 activity.

3. There is not enough evidence to prove that indisulam is a CDK2 inhibitor in the present study. In fact, the results show that palbociclib alone has a more pronounced inhibitory effect on CDK2 (Figure 3E). The relevant issues need to be clarified.

• We believe that indisulam is an indirect CDK2 activity, preventing CDK2 activation and not changing the abundance of the protein. We would therefore not expect to observe a decrease in CDK2 total protein when treating cells with indisulam (Figure 3F). On the other hand, we do observe reduction in active CDK2 in indisulam treated cells (Figure 5N) which indicates that indisulam prevents CDK2 activation. Furthermore, palbociclib inhibits CDK4/6 which is upstream of CDK2 in the cell cycle and has been previously demonstrated to influence CDK2 abundance (PMID: 27020857). It is therefore not unexpected that total levels of CDK2 are reduced in palbociclib treated cells, which we also observed in RNA seq experiment. We added a comment clarifying this in the text (line 171-173).

4. Authors indicated that treating cells with indisulam led to downregulation of cyclin H, which prevented CDK2 activation. However, indisulam did not reduce cyclin H levels till 48 hours of treatment (Figure 5I), but CDK2 activity tests are only analyzed within 24 hours. Such experimental design and results cannot explain that treating cells with indisulam led to downregulation of cyclin H, which prevented CDK2 activation.

We agree that there are differences between the western blot and imaging experiments. The microscopy experiments were performed in an acute setting: a shorter time interval and with much higher concentration of indisulam. It is technically not possible to perform these experiments with longer time points due to effects on cell viability. On the other hand, the Western blot experiment was performed with lower concentration of indisulam which could explain the difference in timing of cyclin H downregulation. Additionally, a small decrease of cyclin H might not be obvious on Western blot, but could already result in lower CDK2 activation. 

5. Authors mentioned that combined treatment with palbociclib and indisulam induced a senescence program and sensitized cells to senolytic therapy. According to the result, this event is only found in A549 cells, but not SUM159 cells (Figure 3D). The relevant issues need to be clarified.

We agree that the effect of ABT-263 in SUM159 was not convincing. We believe the senescence induction was less efficient in that experiment. We therefore repeated the experiment while monitoring senescence induction and now observed sensitisation of senescent cells to ABT263 treatment. We have changed the figure to include new data (Figure 3E).

6. There is no evidence to support that COMBO therapy induces tumor senescence in vivo and resensitizes tumors to senolytic therapy, and relevant markers need to be checked.

Senescence characterisation in vivo is challenging due to heterogeneity of senescence induction in vivo. As the cell line we used for the xenograft model (A549) is p16 deficient we chose to characterise p21 and Ki67 instead. We agree that sensitivity to senolytic therapy in vivo would be interesting, but we believe this experiment is out of the scope of this project.

---

## [Decision Letter · Decision Letter 1]

6 Jul 2022

PONE-D-21-35335R1Indisulam synergizes with palbociclib to induce senescence through inhibition of CDK2 kinase activityPLOS ONE

Dear Dr. Leite de Oliveira,

Thank you for submitting your manuscript to PLOS ONE. After careful consideration, we feel that it has merit but does not fully meet PLOS ONE’s publication criteria as it currently stands. Therefore, we invite you to submit a revised version of the manuscript that addresses the points raised during the review process.

As pointed out by reviewers, the synergy of the drug combination needs to be determined quantitatively and the statistical consideration was lacking for some experiments. 

We look forward to receiving your revised manuscript.

Kind regards,

Wei Xu

Academic Editor

PLOS ONE

Reviewers' comments:

Reviewer's Responses to Questions

**Comments to the Author**

1. If the authors have adequately addressed your comments raised in a previous round of review and you feel that this manuscript is now acceptable for publication, you may indicate that here to bypass the “Comments to the Author” section, enter your conflict of interest statement in the “Confidential to Editor” section, and submit your "Accept" recommendation.

Reviewer #3: (No Response)

Reviewer #4: (No Response)

2. Is the manuscript technically sound, and do the data support the conclusions?

Reviewer #3: Yes

Reviewer #4: Yes

3. Has the statistical analysis been performed appropriately and rigorously? 

Reviewer #3: Yes

Reviewer #4: No

4. Have the authors made all data underlying the findings in their manuscript fully available?

Reviewer #3: Yes

Reviewer #4: Yes

5. Is the manuscript presented in an intelligible fashion and written in standard English?

Reviewer #3: Yes

Reviewer #4: Yes

6. Review Comments to the Author

Reviewer #3: In this manuscript by Ziva Pogacar et al., the authors tried to broaden clinical utility of CDK4/6 inhibitor in some cancers that CDK4/6 inhibitors are limited used. They performed functional shRNA screens in palbociclib resistant cells and found knockdown of CDK2 makes the cells more sensitive to palbociclib. The experiments are performed in multiple cell lines and also confirmed in vivo, overall, the experimental designs are rigor, the conclusions are supported by solid data. There are a few comments needed to be addressed:

1. Indisulam was used as a CDK2 inhibitor in this paper, so I prefer the author show some background of indisulam that used as CDK2 inhibitor in the introduction part.

2. Fig. 1C, is the CAL-51-CKD2 knockout cell line monoclonal cell line or polyclonal cell line, why there are still bands in the two CAL-51-CKD2 knockout cell lines.

3. For one of reviewers’ questions: There is no direct evidence to prove that indisulam synergizes with palbociclib to induce senescence through inhibition of CDK2 kinase activity (Title). The author did not present evidence that indisulam inhibits CDK2 kinase activity. According to the author’s response, the author didn’t provide direct evidence to confirm CDK2 kinase activity was prevented by indisulam and Palbociclib. To address this question, the author should at least test the expression of phosphorylation of CDK2 after treated with indisulam or combine with palbociclib.

Reviewer #4: In this manuscript, the author found that indisulam prevented CDK2 activation and can synergistically induce senescence when it was combined with Palbociclib, a CDK4/6 inhibitor. I have several comments for the author to consider.

1. The author declared a synergy between indisulam and palbociclib in the induction of senescence,

but no related data to show this. The author may consider using combination index (CI) values

or other similar indices to exhibit synergistic effects for related detections (Control, PALBO, IND, COMBO).

2. It would be better if the authors had data for senolytic drug ABT-263 assays with colorectal cells.

3. For detections with PALBO, IND, and COMBO, the author may consider applying 2-factor ANOVA or 3-factor ANOVA (if having a time effect) analysis.

4. For Fig.3D and similar figure panels in other figures, the authors only compared all other groups to

the combination groups. Please clarify or modify.

5. It would be better if the author could quantify protein expression for Fig.3F and 5J by looking at the loading controls. The changing trends will be easy to distinguish accordingly.

6. Please add statistical significance for Figures S4A and B

7. The authors only provide statistical significance for the endpoint in Figure 4A. Please modify.

8. For Figures 5C, 5D, 5F, 5H, 5J, 5K, and 5N, the authors frequently changed the concentrations of PALBO and/or IND used in different detections compared to previous assays. Please clarify each of them.

9. The data is not strong enough to support that indisulam-induced CCNH downregulation prevents CDK2 activation through the CAK-complex inactivation. Please consider providing more explanations in the discussion section.

10. In the legend of Figure S1, does “n = 2” means two biological replicates, or was the experiment repeated twice? It is confusing. Please check all legends for similar cases.

11. Please check if one-way ANOVA or two-way ANOVA analysis were used correctly. For example, Figure 1G, Figure 3D, etc.

12. The authors may consider providing an extra figure to better summarize mechanisms or regulatory pathways found.

13. line 166-173, please cite figure panels appropriately.

7. PLOS authors have the option to publish the peer review history of their article (what does this mean?). If published, this will include your full peer review and any attached files.

Reviewer #3: No

Reviewer #4: No

---

## [Author Response · Author response to Decision Letter 1]

29 Jul 2022

We would like to thank the reviewers for their careful reading of our manuscript and constructive comments. We believe that this second round of revisions further improved the quality of the manuscript and we hope you agree that we adequately addressed the reviewer’s comments. 

REBUTTAL PLOS ONE

Reviewer #3: In this manuscript by Ziva Pogacar et al., the authors tried to broaden clinical utility of CDK4/6 inhibitor in some cancers that CDK4/6 inhibitors are limited used. They performed functional shRNA screens in palbociclib resistant cells and found knockdown of CDK2 makes the cells more sensitive to palbociclib. The experiments are performed in multiple cell lines and also confirmed in vivo, overall, the experimental designs are rigor, the conclusions are supported by solid data. There are a few comments needed to be addressed:

We were happy to hear the reviewer agrees that our experiments support the conclusions and our findings are relevant to the field. 

1. Indisulam was used as a CDK2 inhibitor in this paper, so I prefer the author show some background of indisulam that used as CDK2 inhibitor in the introduction part.

We agree with the reviewer that it is important to introduce previous literature on indisulam as a CDK2 inhibitor. We added this information and a citation to the introduction (line 88-89). 

2. Fig. 1C, is the CAL-51-CKD2 knockout cell line monoclonal cell line or polyclonal cell line, why there are still bands in the two CAL-51-CKD2 knockout cell lines.

Those cell lines are indeed polyclonal, which is why there is some background CDK2 expression visible on the Western blot. We have clarified that the lines are polyclonal in the figure legend (line 382).

3. For one of reviewers’ questions: There is no direct evidence to prove that indisulam synergizes with palbociclib to induce senescence through inhibition of CDK2 kinase activity (Title). The author did not present evidence that indisulam inhibits CDK2 kinase activity. According to the author’s response, the author didn’t provide direct evidence to confirm CDK2 kinase activity was prevented by indisulam and Palbociclib. To address this question, the author should at least test the expression of phosphorylation of CDK2 after treated with indisulam or combine with palbociclib.

We agree with the reviewer that the data on CDK2 activity is indirect. However, our results on CDK2 reporter indicate that indisulam prevents activation of CDK2. The experiment assessing endogenous CDK2 phosphorylation on Thr160 (activation) is unfortunately technically not feasible, because the antibody is not specific to CDK2 but also recognises CDK1 phosphorylation on Thr161. 

Reviewer #4: In this manuscript, the author found that indisulam prevented CDK2 activation and can synergistically induce senescence when it was combined with Palbociclib, a CDK4/6 inhibitor. I have several comments for the author to consider.

1. The author declared a synergy between indisulam and palbociclib in the induction of senescence, but no related data to show this. The author may consider using combination index (CI) values or other similar indices to exhibit synergistic effects for related detections (Control, PALBO, IND, COMBO).

We agree with the reviewer that combination indexes might be useful when showing synergistic effects on cell viability of two drugs. However, in our case the phenotype we observe upon combination is senescence not cell death. Senescent cells are known to change their metabolism so a metabolic read-out (such as resazurin assay) typically used to calculate combination indices is not reliable in this case. We therefore opted for proliferation assays to show the effect of combination treatment to cell growth and show full growth arrest upon combination treatment, while cells grow out in individual treatments (Figure 3B and S3B). 

2. It would be better if the authors had data for senolytic drug ABT-263 assays with colorectal cells.

We agree with the reviewer that it would be beneficial to show response to ABT-263 in additional cell lines. We have added data on DLD-1 and RKO parental and senescent cells treated with ABT-263 to Supplemental figure 3E.

3. For detections with PALBO, IND, and COMBO, the author may consider applying 2-factor ANOVA or 3-factor ANOVA (if having a time effect) analysis.

Thank you for your comment. To analyze the proliferation experiments we used 2-factor ANOVA for genetic experiments (Figure 1E, 2C, 2H and S1D) where one factor is genotype and another is treatment. For drug treatment experiments we used 1-factor ANOVA, as we only have one factor: treatment and we analyzed every cell line separately. We chose to only analyze the end points of the proliferation assay to allow easier interpretation together with colony formation assays and SA-Bgal stainings which are both end-point assays. 

4. For Fig.3D and similar figure panels in other figures, the authors only compared all other groups to the combination groups. Please clarify or modify.

We agree with the reviewer that the comparison to combination group should be further explained. As palbociclib is already known to be a weak senescence inducer, we believe that comparing each treatment to untreated control would not be so informative. We aimed to compare the individual treatments to the combination group to show that combination is superior in senescence induction, which is why we compared each treatment group to the combination. We have clarified the choice of analysis in the methods section (line 857-859).

5. It would be better if the author could quantify protein expression for Fig.3F and 5J by looking at the loading controls. The changing trends will be easy to distinguish accordingly.

We agree that the changes in protein abundance are sometimes not very obvious. We have therefore quantified the protein abundance as suggested by the reviewer and added them to the revised manuscript and figures (see Figure 3G and Supplemental Figure 5B).

6. Please add statistical significance for Figures S4A and B

We agree with the reviewer that statistical analysis would help the interpretation of the data in Figure S4A and B. We performed the analysis and added them to the revised manuscript (Figure S4A and B). 

7. The authors only provide statistical significance for the endpoint in Figure 4A. Please modify.

The reviewer is correct that we analysed tumor volumes at the end point of the experiment. We decided for that as samples for stainings were taken at the same time point, so we believe this allows for more relevant conclusions and comparisons between different groups. 

8. For Figures 5C, 5D, 5F, 5H, 5J, 5K, and 5N, the authors frequently changed the concentrations of PALBO and/or IND used in different detections compared to previous assays. Please clarify each of them.

The reviewer is correct that we used different concentrations in different experiments. We typically used lower concentrations for long term experiments, such as proliferation assays, colony formation assays and B-gal stainings (Figure 5C, D, N) and higher concentrations for short term experiments such as imaging experiments with CKD2 reporter, short-term Western blots and FACS (Figure 5G, H, I, J). We clarified the use of concentrations between different experiments in the results section of the revised manuscript (line 234-235). 

9. The data is not strong enough to support that indisulam-induced CCNH downregulation prevents CDK2 activation through the CAK-complex inactivation. Please consider providing more explanations in the discussion section.

We agree with the reviewer that other factors might play a role in senescence induction. We added an additional explanation to the discussion (lines 326-328). 

10. In the legend of Figure S1, does “n = 2” means two biological replicates, or was the experiment repeated twice? It is confusing. Please check all legends for similar cases.

We agree with the reviewer that the figure legends could be confusing. We have adapted all figure legends to clarify independent experiments and technical replicates. 

11. Please check if one-way ANOVA or two-way ANOVA analysis were used correctly. For example, Figure 1G, Figure 3D, etc.

This comment is similar to comment no.3. As explained above, we performed two-way ANOVA when analysing genetic experiments (one factor is genotype, another is treatment) - for example Figure 1G, S1F, 2E, 2J and one way ANOVA for drug experiments (one factor is treatment) for example Figure 3D, S3D. 

12. The authors may consider providing an extra figure to better summarize mechanisms or regulatory pathways found.

We agree with the reviewer that a model can clarify the pathways involved. We have added a model to Figure 5O. 

13. line 166-173, please cite figure panels appropriately.

We have re-checked and adjusted citing of the figures where needed.

---

## [Decision Letter · Decision Letter 2]

4 Aug 2022

Indisulam synergizes with palbociclib to induce senescence through inhibition of CDK2 kinase activity

PONE-D-21-35335R2

Dear Dr. Oliveira

We’re pleased to inform you that your manuscript has been judged scientifically suitable for publication and will be formally accepted for publication once it meets all outstanding technical requirements.

Kind regards,

Wei Xu

Academic Editor

PLOS ONE

Additional Editor Comments (optional):

Reviewers' comments:

Reviewer's Responses to Questions

**Comments to the Author**

1. If the authors have adequately addressed your comments raised in a previous round of review and you feel that this manuscript is now acceptable for publication, you may indicate that here to bypass the “Comments to the Author” section, enter your conflict of interest statement in the “Confidential to Editor” section, and submit your "Accept" recommendation.

Reviewer #3: All comments have been addressed

Reviewer #4: All comments have been addressed

2. Is the manuscript technically sound, and do the data support the conclusions?

Reviewer #3: Yes

Reviewer #4: Yes

3. Has the statistical analysis been performed appropriately and rigorously? 

Reviewer #3: Yes

Reviewer #4: Yes

4. Have the authors made all data underlying the findings in their manuscript fully available?

Reviewer #3: Yes

Reviewer #4: Yes

5. Is the manuscript presented in an intelligible fashion and written in standard English?

Reviewer #3: Yes

Reviewer #4: Yes

6. Review Comments to the Author

Reviewer #3: The authors have answered all my questions and all my comments have been addressed. I agree with this manuscript to publish on PLOS ONE.

Reviewer #4: (No Response)

7. PLOS authors have the option to publish the peer review history of their article (what does this mean?). If published, this will include your full peer review and any attached files.

Reviewer #3: No

Reviewer #4: No

---

## [Editor Report · Acceptance letter]

26 Aug 2022

PONE-D-21-35335R2 

Indisulam synergizes with palbociclib to induce senescence through inhibition of CDK2 kinase activity 

Dear Dr. Leite de Oliveira:

I'm pleased to inform you that your manuscript has been deemed suitable for publication in PLOS ONE. Congratulations! Your manuscript is now with our production department. 

Kind regards, 

on behalf of

Dr. Wei Xu 

Academic Editor

PLOS ONE